
# Classification of aerosol population type and cloud condensation nuclei properties in a coastal California littoral environment using an unsupervised cluster model

Samuel A. Atwood[1], Sonia M. Kreidenweis[1], Paul J. DeMott[1], Markus D. Petters[2], Gavin C. Cornwell[3], Andrew C. Martin[4], Kathryn A. Moore[1,3]

[1]Department of Atmospheric Science, Colorado State University, Fort Collins, CO 80523, USA
[2]Department of Marine, Earth and Atmospheric Sciences, North Carolina State University, Raleigh, NC 27695, USA
[3]Department of Chemistry and Biochemistry, University of California San Diego, La Jolla, CA, USA
[4]Climate Atmospheric Science and Physical Oceanography, Scripps Institution of Oceanography, La Jolla, CA, USA

*Correspondence to*: Sonia M. Kreidenweis (soniak@colostate.edu)

**Abstract.** Aerosol particle and cloud condensation nuclei (CCN) measurements from a littoral location on the northern coast of California at Bodega Bay Marine Laboratory (BML) are presented for approximately six weeks of observations during the CalWater-2015 field campaign. A combination of aerosol microphysical and meteorological parameters was used to classify variability in the properties of the BML surface aerosol using a K-means cluster model. Eight aerosol population types were identified that were associated with a range of impacts from both marine and terrestrial sources. Average measured total particle number concentrations, size distributions, hygroscopicities, and activated fraction spectra between 0.08% and 1.1% supersaturation are given for each of the identified aerosol population types, along with meteorological observations and transport pathways during time periods associated with each type. Five terrestrially influenced aerosol population types represented different degrees of aging of the continental outflow from the coast and interior of California and their appearance at the BML site was often linked to changes in wind direction and transport pathway. In particular, distinct aerosol populations, associated with diurnal variations in source region induced by land/sea-breeze shifts, were classified by the clustering technique. A terrestrial type representing fresh emissions, and/or a recent new particle formation event, occurred in approximately 10% of the observations. Over the entire study period, three marine influenced population types were identified that typically occurred when the regular diurnal land/sea-breeze cycle collapsed and BML was continuously ventilated by air masses from marine regions for multiple days. These marine types differed from each other primarily in the degree of cloud processing evident in the size distributions, and in the presence of an additional large-particle mode for the type associated with the highest wind speeds. One of the marine types was associated with a multi-day period during which an atmospheric river made landfall at BML. The generally higher total particle number concentrations but lower activated fractions of four of the terrestrial types yielded similar CCN number concentrations to two of the marine types for supersaturations below about 0.4%. Despite quite different activated fraction spectra, the two remaining marine and terrestrial types had CCN spectral number concentrations very similar to each other, due in part to higher number concentrations associated with the terrestrial type.





## 1   Introduction

Atmospheric rivers (ARs) are tropical moisture advection phenomena that can account for large fractions of the wintertime precipitation in California (Ralph et al., 2004; Dettinger et al., 2011). The winter-spring 2015 CalWater-2015 study (Ralph et al., 2015), and coordinated ACAPEX study (Leung, 2016) that included aircraft- and ship-based observations in the same region, were designed to probe the atmospheric conditions in and around ARs, and to provide new observations of the characteristics of regional aerosols that may interact with these atmospheric moisture features and thereby influence the downwind formation of precipitation. As part of the CalWater-2015 study, ground-based aerosol observations were conducted at the Bodega Bay Marine Laboratory (BML), a coastal California site that is suitable for observation of aerosols in landfalling marine air masses, and in mixtures of marine and continental air.

In marine regions impacted by continental outflow, aerosol chemical and microphysical properties, including particle number concentrations and size distributions, are often moderated by impacts from terrestrial sources (Nair et al., 2013; Wex et al., 2016; Zhao et al., 2016; Phillips et al., 2018). For example, freshly emitted sea spray aerosol particles comprise a mixture of salts with generally high hygroscopicities ($\kappa \sim 0.6$–$1.2$), and co-emitted organic species with lower hygroscopicities ($\kappa \sim 0$–$0.3$) (Prather et al., 2013; Quinn et al., 2014)—classified using the $\kappa$ hygroscopicity parameter (Petters and Kreidenweis, 2007). Bulk hygroscopicity values above 1 are infrequently observed, but have been reported for some background and precipitation impacted marine aerosol populations (Good et al., 2010; Prather et al., 2013). As the aerosol ages, chlorine replacement by uptake of acidic gases can reduce the hygroscopicity of the salts. Aging and organic components result in reported distribution-averaged $\kappa$ values in the range of 0.4 to 0.7 for more pristine or background marine aerosol populations (Keene et al., 2007; Bates et al., 2012; Prather et al., 2013; Quinn et al., 2014; Zhang et al., 2014; Forestieri et al., 2016; Atwood et al., 2017; Royalty et al., 2017; Phillips et al., 2018). In contrast, average hygroscopicities for continental organic aerosol populations are typically in the range of 0.1 to 0.3 due to heavy influence of organic and insoluble material, although individual inorganic sulfates and nitrate salts have values in the same range as many remote marine aerosol populations (Andreae and Rosenfeld, 2008). The cloud-droplet-nucleating activity of particles in coastal regions therefore depends on the composition, size distribution, and degree of aging of the marine particles, along with the characteristics of particles from non-marine sources that become mixed with the marine aerosol. Thus, observation and analysis of the coastal California aerosol environment will support efforts to better understand implications of aerosol properties on precipitation and cloud development in the region, including aerosol influences on how precipitation develops in landfalling ARs in this region.

In this study, observations from the BML ground site during the CalWater-2015 field campaign were used to evaluate the nature and variability of surface (marine boundary layer, MBL) aerosol populations at BML. Observations were classified into periods of similar aerosol and meteorological characteristics using an unsupervised cluster model (Atwood et al., 2017) to derive distinct littoral aerosol population types and link them to source regions. The clusters were assessed with respect to their aerosol characteristics, particularly the contributions of each population type to cloud condensation nuclei (CCN) concentrations that influence the microphysical properties of liquid-phase clouds and the evolution of precipitation in mixed-



phase clouds (e.g., Rosenfeld et al., 2008). In prior work analyzing data from the CalWater-2015 / ACAPEX field experiments, Leung (2016) discusses an AR event that was first observed and sampled from aircraft platforms off the coast of California on 5 February 2015, during the observational period analyzed here. The AR made landfall near BML on the morning of 6 February, and the AR-associated front, producing heavy precipitation along its trajectory, reached the Central Valley and Sierra Nevada in the afternoon. We thus also contrast the aerosol cluster types identified at BML during AR and marine aerosol dominated events to observations outside those periods.

## 2 Methods and site description

Measurements occurred at Bodega Bay Marine Laboratory (BML; 38.32° N, 123.07° W) and took place between January and March 2015. Data utilized in this study were gathered between 23 January and 5 March, when relevant instruments were operational. All aerosol measurements used here were made from within the Colorado State University/National Park Service mobile lab, located approximately 200 m from the coast, and approximately 150 m north of the main BML lab building. Air was sampled through several ½" stainless steel tube inlets routed through the roof of the mobile lab to a height of approximately 5 m, before the flow was split and sent to various instruments described further below. A map and additional description of the site and instrumentation are provided in Martin et al., (2017).

### 2.1 Meteorological data

Local meteorological measurements were obtained from a 10 m surface met tower located approximately 100 m north of the mobile lab, and operated as part of a NOAA/ESRL observation network (White et al., 2013). Air mass source regions were defined using 24-hour backtrajectories that were initiated every three hours during the study period and computed using the HYSPLIT version 4.9 Lagrangian parcel model (Draxler and Hess, 1997, 1998; Draxler et al., 1999; Stein et al., 2015) with the 40 km x 40 km Edas40 meteorological dataset. All trajectories were generated with 100 m arrival heights at BML to characterize transport in the MBL.

### 2.2 Aerosol size distribution measurements (SMPS)

Aerosol size distribution measurements made during CalWater-2015 at BML are described in detail in Martin et al., (2017). This study used size distribution measurements for particle diameters between 14 and 730 nm from a TSI 3080 SMPS using a 0.3 L min$^{-1}$ sample flow rate and 3.0 L min$^{-1}$ sheath flow rate. SMPS scans were conducted approximately every 5 minutes and were subsequently averaged to approximately 15-minute time periods to match additional coincident measurements. The presence of large numbers of particles smaller than approximately 30 nm in diameter was intermittently observed in the size distribution measurements. These particles were likely from a combination of local sources that included vehicle and other activity at BML, local camp and brush fires, and emissions from the nearby town of Bodega Bay, located to the east of the site. This nucleation mode was generally superimposed on more stable aerosol populations, and thus rather than removing the



entire observation during these events, we removed only the contamination mode, as described below and in further depth in
the supplemental material.
The best-fit modal parameterization for each size distribution spectrum was first assessed using a lognormal mixture
distribution fitting algorithm based on Hussein et al., (2005), and as described in Atwood et al., (2017). The algorithm selects
between one and three modes to best represent the size distribution based on empirical rules and maximum-likelihood fitting
criteria, and defines each mode by three parameters (median diameter, geometric standard deviation, and fractional number
concentration). Within each of the size spectra, a fitted mode was identified as local contamination if the following criteria
were met:
• The combined fit's distribution was identified to have more than one mode
• The median diameter of the smallest fitted mode was at or below 20 nm
• The smallest fitted mode's $dN/dlog_{10}D_p$ value was less than 50% of the combined fit's $dN/dlog_{10}D_p$ value at 50 nm
(indicating only a small contribution to total particle count in larger fit modes)
• The smallest fitted mode did not persist continuously for more than three hours
In the case that an observed size distribution spectrum passed all these criteria, the smallest mode was classified as a
contamination mode likely associated with local sources as described above and therefore not representative of the regional
aerosol. These modes were removed from the size distributions and total number concentrations while retaining the remaining
fitted modes. This was accomplished by multiplying the observed $dN/dlog_{10}D_p$ value by one minus the contamination mode's
number fraction of the total distribution for each size bin. An example of the removal is shown in Fig. 1, with an observed
distribution without a local mode shown in (a). Approximately one hour later, a distinct small mode was observed with a
median diameter less than 14 nm and nearly no contribution to particle concentration at diameters above 20 nm (b), while total
number concentration showed a rapid increase (not shown). This mode was no longer present approximately 1.5 hours later.
Therefore, the observation passed all four criteria to be considered local contamination and was removed from the reported
size distribution and total number concentration to give the final corrected distribution (c).
**2.3  CCN measurements**
As described in Martin et al. (2017), size-resolved CCN concentrations (srCCN) were measured using a DMT cloud
condensation nuclei counter (CCN-100) coupled with a TSI 3080 SMPS to scan across a range of particle diameters (12-540
nm at water supersaturation of $s$ = 0.1%, 0.19%, 0.28%, 0.44%, 0.58%, 0.67%). A distribution-averaged apparent
hygroscopicity parameter κ (Petters and Kreidenweis, 2007) was calculated from these scans as described in Petters and
Petters, (2016).
A separate scanning flow CCN system (sfCCN) was used to measure total CCN number concentration as supersaturation was
ramped between approximately 0.08% and 1.1% supersaturation. The system used a DMT CCN-100 instrument that had been
modified by connecting a voltage modulated proportional flow valve to the bottom of the column to control flow (Suda et al.,
2014). The flow rate through the CCN column was increased from 0.2 to 1.2 lpm over 5 minutes while holding the temperature





gradient constant, thereby scanning peak column supersaturation as a function of flow rate and column temperature gradient (Moore and Nenes, 2009). A TSI 3010 Condensation Particle Counter (CPC) was placed in parallel to the CCN to measure total particle number concentration (CN). CCN and CN concentrations were recorded at a frequency of one hertz. Each approximately 5-minute scan was repeated three times, after which the temperature gradient was changed to scan a different range of supersaturations. As the column temperatures took approximately 3 minutes to stabilize, the first scan of each three-repetition set was not analyzed. As the residence time in the sfCCN varied with total flow rate when compared to the parallel CPC, the CPC timestamps were empirically adjusted to ensure the 1 Hz data points from both instruments were aligned, as described in more detail in the Supplemental Material. Calibrations of both CCN systems were conducted following the methodology described in Suda et al., (2012).

Example sfCCN concentrations and activated fraction spectra are shown in Fig. 2(a&b), respectively, for the same time stamp as the size distribution data in Fig. 1. The range of CN concentrations as measured by the parallel CPC during this period are shown as red data points (placed for comparison to CCN concentrations at 1.3% supersaturation). During the contamination period approximately one hour later (Fig. 2(c&d)) the effect of the small mode noted in Fig. 1(b) was seen via increased CN concentrations (reaching as high as 5000 cm$^{-3}$) and decreased activated fractions above approximately 0.1% supersaturation. Observed CCN concentrations remained relatively constant indicating the additional particles were too small to activate below 1.1% supersaturation. After removal of this small contamination mode the corrected CN concentrations and activation spectrum Fig. 2(e&f) were similar in characteristics to the earlier non-contaminated period.

## 2.4    Classification of aerosol population type

Classification of aerosol population types impacting BML was conducted using an unsupervised K-means cluster analysis. Such clustering methods utilize properties of the aerosol and environment to identify periods of potentially similar impacts and aerosol population types (Wilks, 2011). Cluster analyses have been used to classify aerosol particle size distributions (Tunved et al., 2004), associate them with various environmental and atmospheric processes (Charron et al., 2008; Beddows et al., 2009; Wegner et al., 2012), and conduct aerosol source apportionment studies (Salimi et al., 2014). Information on aerosol chemistry or composition has also been used for clustering purposes (Frossard et al., 2014), and has been integrated with size distribution measurements and atmospheric transport data to produce cluster results based on multiple types of observations (Charron et al., 2008; Atwood et al., 2017).

The K-means clustering methodology involved selection of specific variables that partially defined the state of the aerosol and meteorological environment at the sampling site. The degree of similarity of the state of the environment between any two data points (i.e. specific times) was estimated by the use of a distance function that grouped data points into clusters that had broadly similar values among the input variables. Here, we utilized the *cluster.KMeans* class of the Python scikit-learn package (Pedregosa et al., 2011) to perform the analysis. Selection of the appropriate number of clusters followed a similar methodology to that described in Atwood et al., (2017). A hierarchical cluster analysis was first created using the *cluster.AgglomerativeClustering* class of scikit-learn to identify potential numbers of clusters using a dendrogram. Various





internal validity measures of the K-means clusters (Beddows et al., 2009; Baarsch and Celebi, 2012) were then used, along
with verification that the results maintained physically distinct and temporally coherent clusters, in order to select the
appropriate number of K-means clusters.
**2.4.1    Cluster variables**
Variables used in this analysis included measurements of aerosol microphysical properties and of meteorological parameters
at BML. Aerosol property variables included the normalized size distribution at each time stamp (normalized to an integrated
value of 1 cm$^{-3}$ by dividing by the total particle number concentration), after correction for local contamination. The
distributions were discretized into 20 logarithmically spaced bins that served as separate variables in the clustering distance
function (e.g. Charron et al. (2008)). Activated fraction spectra for each time stamp were divided into 20 linearly spaced bins
distributed between 0% and 1.1% supersaturation to incorporate CCN properties into the analysis. Total particle number
concentration was included as a separate cluster variable.
Meteorological parameters at BML included the local 10m observed wind velocity as perpendicular u and v component
variables. Additionally, HYSPLIT backtrajectories were assigned to data points closest in time to each trajectory. The
backtrajectory was converted to separate variables for the distance function by determining the distance from the receptor,
initial bearing, and altitude, every three hours backwards along the trajectory for 24 hours, yielding a total of 24 trajectory
clustering variables for each time stamp.
**2.4.2    Distance function**
The Karl Pearson Euclidean distance function (Wilks, 2011) used in the cluster model was modified to include a relative
weight parameter for each input variable as:

$$d_{i,j} = \left[ \sum_{k=1}^{K} w_k \left( x_{i,k} - x_{j,k} \right)^2 \right]^{\frac{1}{2}}$$

(1)

where $d_{i,j}$ gives the Euclidean distance between two data point vectors, $x_i$ and $x_j$, in a $K$-dimensional space (i.e. $K$ nominally
independently measured or observed, orthogonal variables at each time stamp), for each variable, $k$, with relative weight, $w_k$.
Each variable was first standardized, with missing data imputed to values of zero to minimize their impact on the distance
function.
The relative weights for each variable were included to prevent properties of the aerosol from becoming over-weighted in the
cluster model due to having more variables describing them. The size distribution and activated fraction variables, each of
which had 20 variables, were given relative weights of 1/20 such that their total relative weights summed to 1. As these
variables were of primary importance to aerosol population microphysical properties, the other variable groups were decreased
in relative importance. The backtrajectory and wind vector groups were each assigned a relative weight of 0.5, and the total
particle number concentration variable assigned a relative weight of 0.1. As cluster analysis is by its nature an exploratory data





analysis technique, these relative weight values were reached by varying their values and assessing the physical interpretability
of the results.

### 3    2.4.3     Number of clusters

Hierarchical clustering and internal validity measures indicated two, six, eight, and twelve clusters were potentially appropriate
for the K-means analysis. Clusters associated with periods of marine aerosol impacts (discussed further in the next section)
became temporally coherent and physically meaningful after the number of clusters was increased to eight.
In the case of the twelve-cluster analysis, several of the clusters were composed of few or even a single data point, indicating
the model had begun to separate outliers into distinct clusters. In addition, several of the temporally consistent clusters were
split, indicating that too many clusters had been selected. All potential cluster numbers between seven and eleven were then
investigated to determine if physically or temporally coherent population types emerged to a greater degree than the eight-
cluster model. As the eight-cluster option was initially identified as potentially appropriate, and the other potential models did
not improve physical interpretability of the results, the eight-cluster model was selected as the most appropriate unsupervised
classification result.

### 14    3     Aerosol population type classification results and discussion

Three of the eight identified clusters were defined as "marine" population types, as backtrajectory data showed evidence of
transport pathways primarily over ocean areas. These marine types, denoted as clusters M1–M3, tended to have lower average
number concentrations (below approximately 1500 cm$^{-3}$), while the terrestrial clusters (T1–T5) had typical averages between
approximately 2000 and 4000 cm$^{-3}$ and were associated with transport from more terrestrial source regions. The exception to
this was cluster T5, which had number concentrations of roughly 1500 cm$^{-3}$, more oceanic transport pathways, and size
distributions with the largest median diameters among the "terrestrial" clusters. Table 1 provides the cluster-averaged number
concentrations, wind velocities, HYSPLIT accumulated precipitation along the 24-hour trajectory, hygroscopicity parameters
from the srCCN system, and the percentage of all measurements associated with each of the 8 identified clusters. Best-fit size
distribution parameters and best-fit CCN-spectrum activated fraction parameters (see Supplemental Material) for each cluster
are provided in Table 2.
Figure 3 presents the study timelines of measured aerosol (a) and meteorological (b) variables, and the corrected (c) and
normalized (d) aerosol size distributions. The fitted modal diameters are superimposed on the normalized size distributions,
revealing periods of stability in the size distributions as well as periods that are highly variable. Time periods associated with
each cluster are shown as colors in the background of the panels.
Changes between identified cluster types often tracked diurnal changes in wind direction associated with the land/sea-breeze
cycle (Fig. 3b; e.g., 28–30 Jan). Several periods occurred during the study during which this diurnal cycle collapsed and BML
was ventilated with air masses from marine regions for several days at a time (starting on 4 Feb, 17 Feb, and 26 Feb). Clusters



M1, M2, and M3 were selected by the model more consistently during these extended periods and confirmed their
characterization as marine aerosol population types.
Normalized size distributions for the eight clusters are shown in Fig. 4, along with the average total corrected particle number
concentrations and the total number of data points included in each cluster. Clusters with terrestrial and/or anthropogenic
influence were ordered by increasing median diameter of the mode with the largest number fraction. HYSPLIT backtrajectories
and wind rose plots for each data point included in each cluster are shown in Fig. 5.

### 3.1    Marine population types

Further analysis of the marine clusters showed generally distinct meteorological conditions associated with each. Cluster M1
primarily occurred toward the end of the study period, during a period of high velocity winds from the northwest.
Backtrajectories agreed with local winds and showed generally faster transport velocities. This cluster dominated during 26
and 27 February, a period during which the cleanest air masses and lowest number concentrations of the study were observed,
reaching as low as 50 cm$^{-3}$ during the height of the event. The normalized size distributions indicated that the Aitken mode
dominated the number distribution (Fig. 4), and also suggested the presence of a somewhat larger mode (particles larger than
~400 nm) that may have been associated with generation of sea spray by the higher wind velocities. Backtrajectories indicated
that airmasses had passed over the ocean to the northwest of BML, while 24-hour accumulated precipitation along the
trajectories (Table 1) indicated rainfall had occurred in the days prior to the airmass' arrival at BML. Each of these findings
was consistent with classification of the M1 cluster as a precipitation scrubbed, clean marine aerosol population.
The marine cluster with the next highest number concentration was M2, with 78% of its total occurrences between 17 and 21
February. The wind rose for this cluster indicated a primarily northwesterly wind, similar to cluster M1 but with much slower
velocities, and backtrajectories with oceanic transport pathways. While HYSPLIT does not always simulate sub-synoptic scale
transport with high fidelity, this pattern may be indicative of slower transport of air from a marine region just off the coast, as
opposed to the direct fast-transport path from more distant ocean regions seen in the M1 type. As observed for M1, the best-
fit average normalized particle size distribution (Fig. 4) was primarily bimodal, consistent with many reports of cloud-
processed background marine aerosol populations (Hoppel et al., 1986; Bates et al., 2000; Wex et al., 2016; Atwood et al.,
2017; Royalty et al., 2017; Phillips et al., 2018), though with a larger accumulation mode number fraction than the M1 type.
Minimal rainfall along the transport pathway was evident in the HYSPLIT accumulated precipitation estimates (Table 1),
indicating no recent precipitation scrubbing and suggesting that more cloud processing without rainout led to larger numbers
of particles in the accumulation mode compared with M1. In contrast to M1, for which a third fitted mode was found above
400 nm, the third mode in M2 occurred at very small particle sizes (diameters less than 30 nm).
The final marine cluster, M3, shared similarities with the other marine types, including a bimodal normalized size distribution
with a minimum near 110 nm and indications of oceanic source regions in local winds and backtrajectories. However, the
average total number concentration was nearly double that of the other marine types. The primary period during which this
cluster occurred was during 4-9 February, bracketing the time before, during, and after landfall of the AR that impacted the



BML region during CalWater-2015. This cluster is therefore interesting as it may be indicative of a unique population type associated with AR meteorological conditions. Some caution is warranted, however, as instrument downtime lead to a gap in the aerosol size distribution dataset during late 6 February through 7 February, during a high-wind and precipitation period that marked the landfall of the AR, and thus some key data that could be used to guide the clustering during this event were missing. When confined to the 5-8 February period noted by Leung (2016) when the AR made landfall at BML, average number concentration was 749 cm$^{-3}$ for time periods associated with the M3 cluster (excluding the data gap on 6-7 February), and 1052 cm$^{-3}$ for the entire 5-8 February AR period (including the intermittent periods classified as terrestrial or anthropogenic aerosol).

Backtrajectories for the M3 cluster indicated source regions from just off the coast of the San Francisco Bay area prior to reaching BML, while HYSPLIT accumulated precipitation along the trajectory was the highest of any cluster (Table 1). Flows associated with AR landfall at the coast can be complex (Neiman et al., 2013), however, emissions from this urban area could potentially have mixed with the relatively low particle number counts that would be expected in a precipitation scrubbed AR air mass as it made landfall, accounting for the elevated (> 1000 cm$^{-3}$) total number (CN) concentrations that persisted for much of the AR, including during the high-wind and heavy precipitation period (Fig. 3b) when SMPS and CCN data were not available. However, local generation of fine-mode sea spray aerosol could also be a factor during high winds.

## 3.2   Terrestrial population types

During periods dominated by diurnal shifts in aerosol and meteorological observations, and for short-duration periods during times associated primarily with marine clusters, the cluster model identified clusters that corresponded to terrestrially influenced populations. In the case of the multi-day events dominated by the M1 and M2 types, these short duration periods were often identified as either T4 or T5, clusters that were notable for having largely monomodal normalized size distributions with median diameters around 100 nm. Occurrences of these cluster types were often associated with a spike in number concentration and changes to either wind direction or wind velocity. Thus, the cluster model was able to identify and separate short duration periods of impacts from terrestrial sources during multi-day marine aerosol conditions at BML.

Several longer periods (28-31 January and 13-16 February) were observed during which populations T4 and T5 regularly alternated in tandem with the diurnal land/sea-breeze shift. Similar diurnal-shift behavior, but between clusters T2 and T3, occurred during 25–28 January and 1–5 March. During these diurnal shifts between various terrestrial clusters, the cluster with the larger median diameter was typically associated with the sea-breeze and transport from oceanic regions, while the smaller diameter cluster was associated with the land-breeze and transport from terrestrial regions. As aging of terrestrial aerosol typically leads to an increase in the median diameter of the aerosol modes, these four clusters may therefore be indicative of various degrees of aging of regional terrestrial aerosol during "sea-breeze resampling" at BML (Martin et al., 2017). Resampling occurs when terrestrial and marine aerosol populations mix and flow across the coastal boundary at low levels, leading to a region of mixed aerosol populations wherein the larger number concentrations associated with terrestrial types dominate the observed number concentrations in the resulting littoral zone airmass. When this diurnal cycle collapsed and





BML was subjected to extended periods of sea-breezes and ventilation by air masses almost exclusively from ocean regions,
the cluster model selected marine cluster types, indicative of marine airmasses that had not experienced much mixing with
terrestrial air masses.
The terrestrial type T1 featured a dominant mode of particles with median diameters around 30 nm, indicating relatively little
aging of the particles had occurred. Both low-level winds and backtrajectories during this cluster type indicated transport
pathways from many directions (Fig. 5), though with the highest wind speeds of the terrestrial clusters, consistent with less
time between the aerosol source and observation at BML. This cluster occurred primarily during two periods, 23–24 February
and 28 February–1 March. During these times, normalized size distributions and modal median diameter fits shown in Fig. 3
indicated that the T1 cluster grew into other clusters with more dominant accumulation mode sizes over the course of several
days. The T1 cluster may therefore identify a freshly emitted population type or a recent new particle formation event.

## 3.3    CCN and activated fraction spectra characteristics

The best-fit activated fraction spectra, as functions of water supersaturation, are shown for all valid data points in each of the
clusters in Fig. 6(a). As a general comparison against other reported values for aerosol activation spectra, Fig. 6 also shows
the parameterized spectra reported by Paramonov et al. (2015) for a range of measurement locations from the European
EUCAARI Network (grey background) and an overall typical average value (black line). The EUCAARI activation spectra
were drawn from a range of marine, littoral, and continental sites, and were impacted by both marine and terrestrial airmasses,
subject to a variety of emissions. The BML spectra spanned much of the range reported for the EUCAARI network, with the
M2 cluster slightly above this range at intermediate superaturations. Activated fraction spectra are independent of total number
concentration, thus the effect of particle size on activation is evident for the BML cases, with clusters with larger size modes
having higher activated fractions across the range of measured supersaturations. These results show the wide range of activated
fraction spectra at BML associated with differences in aerosol population type, and the corresponding complexity of the
population characteristics at this site.
In the CalWater-2015 dataset, marine population types all reached activated fractions of about 0.2 at supersaturations around
0.1% to 0.15%, while the terrestrial types did not reach equivalent fractions until supersaturations between approximately
0.18% and 0.6% were reached. Terrestrial population types with smaller median diameters tended to have less fractional
activation across the full range of measured supersaturations, leading to activated fractions at 1.0% supersaturation that varied
from approximately 0.3 to 0.85. However, due to the generally higher total particle number concentrations, and despite lower
activated fractions of the terrestrial populations, differences in CCN concentrations between marine and terrestrial types (Fig.
6b) were smaller than the differences in the activated fraction spectra. Only at supersaturations above approximately 0.5% did
the CCN concentration for the terrestrial types (except T1, associated with many fresh, small particles) consistently exceed
those of the marine types (Fig. 6b). Between approximately 0.1% and 0.4%, CCN concentrations were often similar between
the marine and terrestrial types.



### 3.4 Comparison of reconstructed and directly-measured CCN spectra

Average values for observations of the hygroscopicity parameter $\kappa$ from the srCCN system are given for each cluster in Table 1. Mean $\kappa$ values for the three marine population types were higher than for any of the terrestrial clusters, with the $\kappa$ for the marine populations found to be significantly different ($p < 0.05$) from those for any of the terrestrial clusters.

Mean measured $\kappa$ values of 0.49 and 0.46 for the M1 and M3 population types, respectively, appeared to be an average of the coastal outflow marine cases presented by Phillips et al., (2018), which had modal values of $\kappa$ between 0.3 and 0.54. Mean values for terrestrial population types varied between 0.15 and 0.25, and were consistent with the 0.2 value for Aitken mode of continental aerosols reported by Phillips et al. However, we note that $\kappa$ values measured in this study and used in the closure calculations were derived from supersaturated CCN measurements, whereas Phillips et al. reported $\kappa$ from humidified growth factors at 80% RH. Prior work has shown that $\kappa$ is not always consistent across this large of a span of RH (Irwin et al., 2010; Whitehead et al., 2014), thus contributing to uncertainty in such comparisons. The hygroscopicity for the final marine M2 type was 0.30, near the lower end of typical values for marine aerosol in regions of continental outflow, but still above those of the terrestrial population types. As the M2 cluster was also the only marine population type with no indication of recent precipitation scrubbing of the airmass prior to arrival at BML (Fig. 3b), some combination of influences from cloud processing, marine, and terrestrial or anthropogenic sources may result in the observed hygroscopicity values between those of the other population types.

The cluster-average hygroscopicities from Table 1 were combined with the average cluster size distributions from Table 2 to create a reconstructed activated fraction spectrum for each cluster. These reconstructions are compared with the direct-measured CCN spectra in Fig. S3 and Table 2. Generally, the reconstructed spectra are within one standard deviation of the directly measured spectra from the sfCCN system. However, the reconstruction overpredicts activated fraction for the marine clusters, with the largest discrepancies at low supersaturations and low CCN number concentrations. Similar behaviors in overprediction of CCN concentrations based on reconstructions using hygroscopicity and size distributions have been noted before (McFiggans et al., 2006). Kammermann et al., (2010) reviewed a number of studies that compared such predicted CCN concentrations against observed values and found biases were often largest at supersaturations below 0.3%, where predictions deviated from observations by factors ranging from 0.6 to 3.3, though with most studies finding overprediction occurred. They attributed the increasing bias at decreasing supersaturations to increased uncertainty in the critical activation diameter and associated CCN number prediction, though they also noted that this discrepancy between predicted and observed CCN concentrations has not been fully resolved. At larger particle diameters measurement uncertainty increases due to imprecise particle size cuts, losses in inlets and tubing, and inversion uncertainties, along with generally lower number concentrations than at smaller particle diameters, leading to higher expected uncertainty in CCN predictions and reconstructions when the critical activation diameter is in this range of particle sizes. At low supersaturations where BML data showed an overprediction bias the critical activation diameter would be above 150 nm. The marine types that had the largest biases at low supersaturations also tended to have larger fractions of particles at these large sizes. This would be expected to add to uncertainty in ways



similar to those noted by Kammermann et al., and potentially explain the larger discrepancies in CCN prediction at lower
supersaturations in the marine aerosol types.
Further investigation of the closure between predicted and observed CCN concentration was conducted using two prediction
models. Hygroscopicity derived predictions using cluster average $\kappa$ and normalized size distribution values were generated
and compared against all observed CCN concentrations by the sfCCN system in Fig. 7(a). Similarly, the predicted CCN
concentration using cluster average activated fraction spectra were compared against observations in Fig. 7(b). Both models
predicted the activated fraction using the cluster type identified during the observation, which was then multiplied by the
observed total number concentration at the observation time. The hygroscopicity and size model showed overprediction
compared to both observations and the activated fraction model (Fig. 7c), with best-fit slopes of 1.08 and 1.09 respectively.
The activated fraction model predictions did improve on the hygroscopicity model, with a slope of 1.00 and $R^2$ values
increasing from 0.87 to 0.89, though this is in part due to the model being based on a direct fit of the observed data.
Nevertheless, the degree of closure between model predictions and observed CCN concentrations is similar to closures
previously reported in field studies (e.g. Bougiatioti et al., 2009; Kammermann et al., 2010).

## 3.5 Aerosol optical properties

While optical properties of the various aerosol population types were not directly measured, a simple optical reconstruction
was conducted to evaluate potential differences between the population types due to differences in size distribution and
hygroscopicity. Average particle size distributions and measured average $\kappa$ values were used to grow particles to equilibrium
with relative humidity across a range of values between 0% and 99%, followed by estimation of mass scattering efficiency on
a dry mass basis using Mie theory (Bohren and Huffman, 1983). For the purposes of this simple optical comparison,
supersaturated $\kappa$ values used for this analysis were assumed to be sufficient to provide an estimate of subsaturated hygroscopic
growth. An assumed dry index of refraction of $1.5 + 0.0i$ and density of $1.0$ g cm$^{-3}$ (e.g. Remer et al., 2006) were used for all
population types in order to estimate the relative effect of differences in size distribution, hygroscopicity, and relative humidity
on scattering properties at a wavelength of 550 nm. The index of refraction of each humidified particle was adjusted based on
volume mixing with water ($m = 1.33 + 0.0i$). Reconstructed mass scattering efficiencies for each of the population types are
shown in Fig. 8.
Computed dry mass scattering efficiencies at 550 nm ranged between 4.2 and 7.7 m$^2$ g$^{-1}$, although actual values would be
expected to be lower due to expected particle dry densities higher than 1 g cm$^{-3}$. Further, these values represent only the
contribution to mass scattering efficiencies from the sub-micron aerosol, whereas super-micron aerosol, including particles
generated by sea spray in littoral environments, can represent a large fraction of the total light scattering. The M1 and M3
marine types, which included the largest fraction of larger accumulation mode particles, had the highest associated dry mass
scattering efficiency. The M2 marine type, which occurred at generally lower wind speeds than the other marine types and
thus had fewer larger particles associated with wind generated sea spray aerosol (O'Dowd and Leeuw, 2007), had a lower dry
mass scattering efficiency than several of the terrestrial types with the largest median particle diameters. However, at relative



humidity values above roughly 60%, as would typically be expected in littoral environments such as BML, the marine
population types all yielded expected mass scattering efficiencies above those of the terrestrial types. For the marine types at
a relative humidity of 95%, the mass scattering efficiencies on a unit-density dry mass basis were between 34 and 49 $m^2\ g^{-1}$,
roughly twice the range of the terrestrial types, 16 to 25 $m^2\ g^{-1}$. As an additional point of comparison, similar reconstructions
with MODIS fine mode ocean aerosol populations (Remer et al., 2006) yielded dry mass scattering efficiencies between 3.4
and 5.4 $m^2\ g^{-1}$, and 33 and 54 $m^2\ g^{-1}$ at 95% RH, indicating the marine types were within the same range as assumptions used
for MODIS marine aerosol populations.

## 4    Summary

The unsupervised cluster model analysis successfully identified distinct aerosol population types in the littoral zone aerosol at
BML during CalWater-2015. The time periods selected by each cluster tended to be both temporally and physically coherent.
Clusters also tended to be grouped into periods with physically meaningful microphysical properties that could be associated
with meteorological processes and expected sources and transport pathways. For example, the clustering methodology
identified regular diurnal swings in aerosol properties associated with land/sea-breeze changes and assigned two distinct,
terrestrially-influenced aerosol types during these periods. Overall, the clustering results for the CalWater-2015 dataset
produced a reliable set of aerosol population types, and appropriately screened intermittent periods of impacts from various
other sources as an outcome of the classification. Both marine and terrestrially influenced aerosol population types were
identified by the unsupervised cluster model. Several marine events that persisted for days were identified as distinct in
character from each other—differing in the degree of cloud processing and precipitation removal prior to arrival at the
measurement site, and in the extent to which high winds contributed larger sea spray particles. About 10% of the observations
were associated with a terrestrial population with a large fraction of small particles, indicating it was affected by relatively
fresh emissions and/or new particle formation.
A primary motivation for CalWater-2015 was improving characterization of the regional aerosol and how it might affect the
formation of precipitation. The CCN activation spectra observed at BML spanned a full range reported in the literature, from
clean marine to strongly terrestrially influenced in character. However, differences in total aerosol number concentrations
associated with the marine and terrestrial types partially offset the differences in activated fraction over a range of measured
supersaturations, such that the variability in CCN concentrations between some marine and terrestrial aerosol types at some
supersaturations was less than expected from the differences in the averaged total aerosol particle concentrations.
In this littoral region sea-breeze resampling and complex mixing between marine, terrestrial, and free-tropospheric air masses
lead to complex aerosol populations. Determination of which aerosol types can become incorporated into AR events and
thereby affect cloud and precipitation over California depends strongly on the individual nature of the flow regimes and
potential aerosol source regions that they tap into. Nevertheless, the observational data set had cases for which marine and
terrestrial CCN concentrations were comparable at supersaturations below approximately 0.4%, and thus changes in aerosol





types would result in little change to initial drop populations in the resulting stratus cloud, except at very low supersaturations.
Thus, mixed-phase microphysical processes occurring in those clouds might also be expected to be similar whether marine or
terrestrial aerosols served as the nuclei for the supercooled droplets. In contrast, for clouds forming at higher maximum
supersaturations, terrestrial aerosol populations are expected to yield higher drop concentrations than marine types, with the
exception of a terrestrial population characterized by primarily small, recently emitted or newly formed particles. Thus the
droplet size distributions formed on terrestrial vs. marine types, for liquid clouds formed in stronger updrafts or at higher
cooling rates that reach such higher supersaturations, should be distinctly different.
At the BML observational site, apparent aerosol types changed over much shorter time scales than a multi-day classification
based on prevailing meteorological conditions would suggest, consistent with complex flows in coastal zones. Inclusion of
both meteorological and aerosol properties into the schemes used in unsupervised cluster models therefore yielded
improvements in classification of aerosol and air mass types observed at the surface, and successfully identified changes at
time resolutions on the order of hours. Further work into classification of aerosol populations in complex or highly variable
regions may therefore benefit from inclusion of a wider range of measurements or observed properties into clustering methods.

**Data availability**

Measurement data for the scanning flow CCN used in this study are available at doi:10.25675/10217/180097
Additional experimental data used in the analysis and presented in the figures are provided in an online data repository at
doi:10.5281/zenodo.XXXXXXX (available upon publication).

**Author Contributions**

Samuel Atwood performed the sfCCN system investigation and data curation, along with conceptualization, methodology,
formal analysis, and writing for this work. Sonia Kreidenweis was involved with writing the manuscript and, along with Paul
DeMott, conducted conceptualization, funding acquisition, administration, and supervision of the project. Markus Petters
performed the srCCN system investigation at BML, along with methodology development for the CCN systems, and formal
analysis for the srCCN results. Andrew Martin assisted with review and editing of the manuscript, and along with Gavin
Cornwell, assisted with investigation and data curation at the BML sampling location. Kathryn Moore performed the SMPS
system investigation and assisted with sfCCN system investigation at BML.

**Acknowledgements**

*This material is based upon research by the Office of Naval Research under Award Number N00014-16-1-2040. This work*
*was supported by NSF award number 1450690 (MDP), NSF award number 1450760 (SAA, SMK, PJD), and NSF award*
*number 1451347 (GCC, ACM, KAM). Assistance in operation of the research site was provided by Nicholas E. Rothfuss, Hans*
*Taylor, Ezra Levin, Christina McCluskey, Yvonne Boose, Gregg Schill, Camille Sultana, and Kim Prather. The assistance by*
*BML staff and Bodega Marine Reserve for lending laboratory space, assisting with measurement site improvements, and*
*permission to measure on Reserve land is gratefully acknowledged. The loan of a CCN instrument by Jeff Reid and the U.S.*
*Naval Research Laboratory, Monterey is also gratefully acknowledged.*



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



# Tables

**Table 1** Aerosol and meteorological parameters for each of the cluster time periods. Cluster mean values are given for total particle number concentration, κ hygroscopicity parameter from the srCCN system, HYSPLIT 24-hour accumulated precipitation along the trajectory, and local wind velocity observations. Best-fit size distribution and activated fraction parameters are shown, with activated fraction parameters pertaining to the fit model given in the supplemental information.

| BML Cluster | Percentage of Observations % (total N) | Total Number Concentration (# cm$^{-3}$) (std dev) | κ Mean (-) (std dev) | Wind Velocity u (m/s) | Wind Velocity v (m/s) | Trajectory 24-hr Accum Precip (mm) (std dev) |
|---|---|---|---|---|---|---|
| M1 | 5% (169) | 592 (435) | 0.49 (0.23) | 6.76 | -4.22 | 0.31 (0.73) |
| M2 | 9% (308) | 774 (462) | 0.30 (0.10) | 2.77 | 1.16 | 0.07 (0.30) |
| M3 | 10% (345) | 1547 (1177) | 0.46 (0.22) | -2.63 | 5.70 | 1.02 (1.76) |
| T1 | 9% (304) | 1975 (2166) | 0.25 (0.17) | 0.25 | 4.12 | 0.51 (1.21) |
| T2 | 17% (566) | 3189 (2055) | 0.18 (0.07) | -1.90 | 2.70 | 0.16 (0.70) |
| T3 | 16% (545) | 3816 (3447) | 0.15 (0.06) | -0.70 | 2.25 | 0.03 (0.21) |
| T4 | 20% (659) | 2306 (1166) | 0.17 (0.06) | 0.46 | 2.49 | 0.01 (0.04) |
| T5 | 14% (461) | 1459 (600) | 0.20 (0.07) | 2.72 | 0.67 | 0.00 (0.00) |

**Table 2** Cluster best-fit size distribution and activated fraction parameters are shown, with activated fraction parameters pertaining to the fit model given in the supplemental information.

| BML Cluster | Avg Size Dist - Mode 1 Median (nm) | Geometric Std Dev | Number Fraction | Avg Size Dist - Mode 2 Median (nm) | Geometric Std Dev | Number Fraction | Avg Size Dist - Mode 3 Median (nm) | Geometric Std Dev | Number Fraction | Activation Specrum Fit Mode 1 a | b | c | Activation Specrum Fit Mode 2 a | b | c |
|---|---|---|---|---|---|---|---|---|---|---|---|---|---|---|---|
| M1 | 45 | 1.77 | 0.69 | 189 | 1.54 | 0.28 | 561 | 1.22 | 0.03 | 0.12 | 1.34 | 0.09 | 0.38 | 2.72 | 0.71 |
| M2 | 21 | 1.46 | 0.07 | 67 | 1.47 | 0.39 | 185 | 1.45 | 0.54 | 0.1399 | 1.5613 | 0.60 | 0.4331 | 1.4379 | 0.30 |
| M3 | 40 | 1.99 | 0.69 | 182 | 1.59 | 0.32 | -- | -- | -- | 0.1246 | 1.299 | 0.09 | 0.266 | 2.8918 | 0.68 |
| T1 | 28 | 1.78 | 0.9 | 144 | 1.51 | 0.10 | -- | -- | -- | 0.1619 | 2.0963 | 0.09 | 1.528 | 3 | 0.57 |
| T2 | 39 | 2 | 0.88 | 170 | 1.49 | 0.12 | -- | -- | -- | 0.4686 | 1.2076 | 0.02 | 0.5111 | 2.8612 | 0.63 |
| T3 | 59 | 1.8 | 0.95 | 211 | 1.42 | 0.04 | -- | -- | -- | 0.4567 | 1.2878 | 0.08 | 0.63 | 2.6407 | 0.70 |
| T4 | 12 | 1.72 | 0.1 | 76 | 2 | 0.90 | -- | -- | -- | 0.38 | 2.3426 | 0.70 | 0.4228 | 1.2607 | 0.08 |
| T5 | 15 | 1.53 | 0.06 | 94 | 1.85 | 0.94 | -- | -- | -- | 0.28 | 2.16 | 0.75 | 0.42 | 1.28 | 0.15 |





## 32  **Figures**

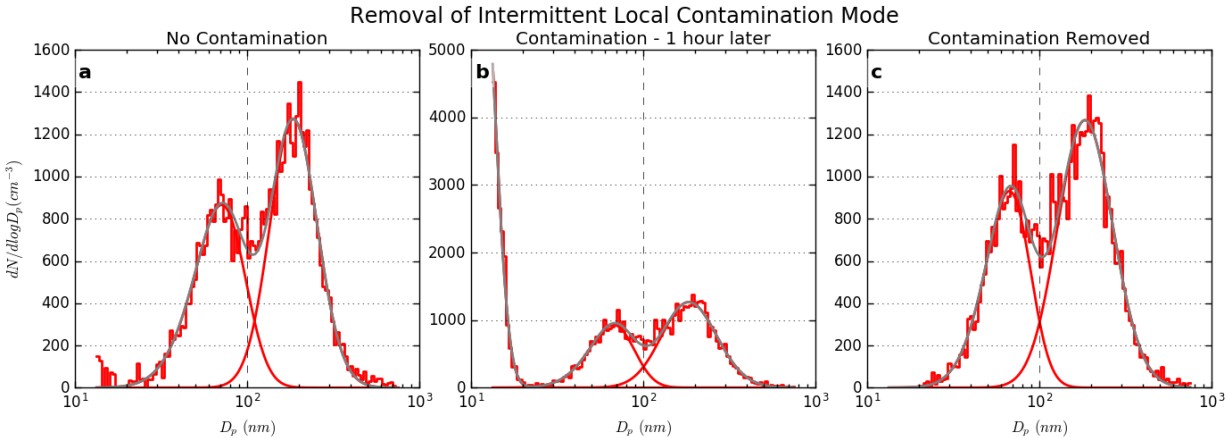


**Figure 1 Example removal of intermittent local contamination mode. (a) An observed size distribution with limited impacts from the smallest nucleation mode particles. (b) An observation approximately 1 hour later with contamination from small particles that are not representative of the regional aerosol. (c) The same size distribution as in (b), but with the smallest contamination mode identified and removed as described in the text.**




**Figure 2  Example spectra measured by the sfCCN system for the same time period as in Figure 1. (a) One second CCN (blue) and CN (red) concentration measurements for a single scan. CCN are shown at the calibrated supersaturations, while CN as measured by the CPC are independent of supersaturation but are shown at 1.3% for comparison. (b) Activated fractions for measured points in (a) along with a best-fit activated fraction spectrum curve (black line). (c) & (d) As in (a) and (b), but for a scan approximately one hour later that was contaminated by an intermittent local ultrafine mode; CN were observed as high as 5000 cm$^{-3}$. (e) and (f) The same spectra as in (c) and (d), but after correction to remove the contamination mode; corrected CN concentrations are now similar to the range in (a).**







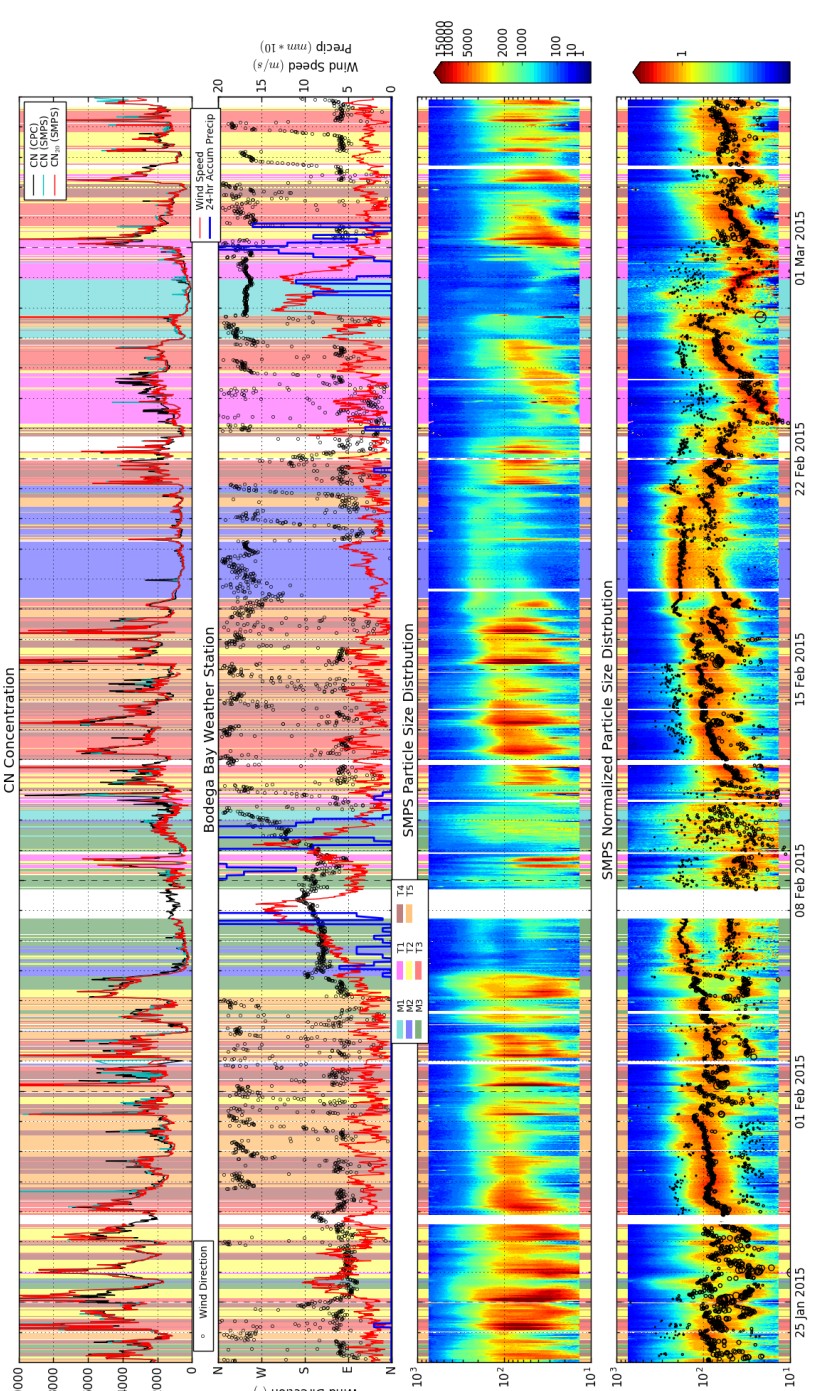

**Figure 3** Timelines of study data; all times shown are UTC. (a) Total particle number concentration at BML measured by the CPC (black), reconstructed from SMPS size distributions (blue), and after correction for removal of local sources (red). (b) BML weather station measurements of wind direction (black circles) and wind speed (red lines), with HYSPLIT 24-hour accumulated precipitation along the airmass trajectory (blue). (c) SMPS size distribution as $dN/dlog_{10}D_P$, and (d) normalized size distributions with best fit median diameters (black circles) with the size of the circle proportional to the number of particles in the mode. Colored backgrounds in each panel are shown for time periods classified as each cluster type for the 8-cluster K-means classification.







**Figure 4  Normalized size distribution for each cluster from the 8 cluster K-means classification. Colored step lines show the average distribution, with best-fit multimodal lognormal fit in black and each mode as colored curves. Observed spectra for each data point in the cluster are shown in grey. Average cluster total particle number concentration and number of data points in each cluster are shown.**





Wind Vector and 100m HYSPLIT backtrajectories





**Figure 5  Meteorological overview for each cluster from the 8 cluster K-means classification. HYSPLIT 24-hr backtrajectories are shown for each time stamp associated with the cluster and colored by the date of arrival at the receptor. Wind rose plots are shown for all BML local 10m wind direction and velocity measurements for each time stamp associated with the cluster. Mean values for 24-hour accumulated precipitation along the trajectory are included in the lower left for each cluster.**

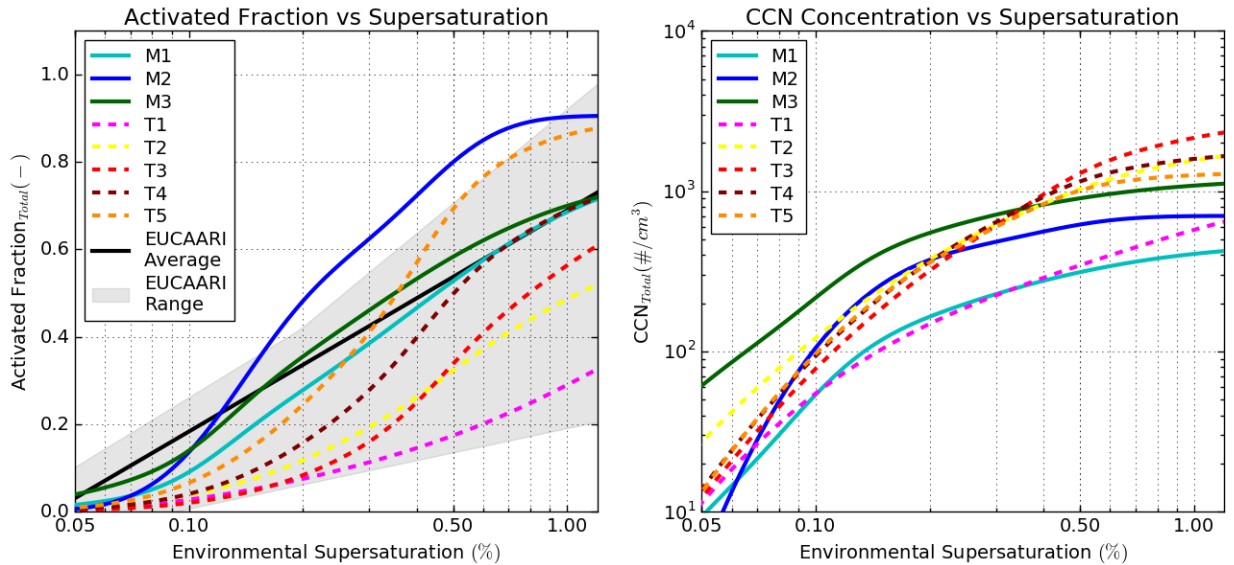

**Figure 6  (a) Best-fit activated fraction spectra for each cluster in the 8 cluster K-means analysis, with marine aerosol population types plotted as solid lines, and terrestrial clusters types as dotted lines. Cluster average spectra are compared against the maximum and minimum values (grey shading) and overall average (black line) reported by Paramonov et al., (2015) from a range of sampling locations in the European EUCAARI network. (b) Parameterized mean CCN concentrations across the same range of supersaturation values for each cluster in the 8 cluster K-means analysis.**





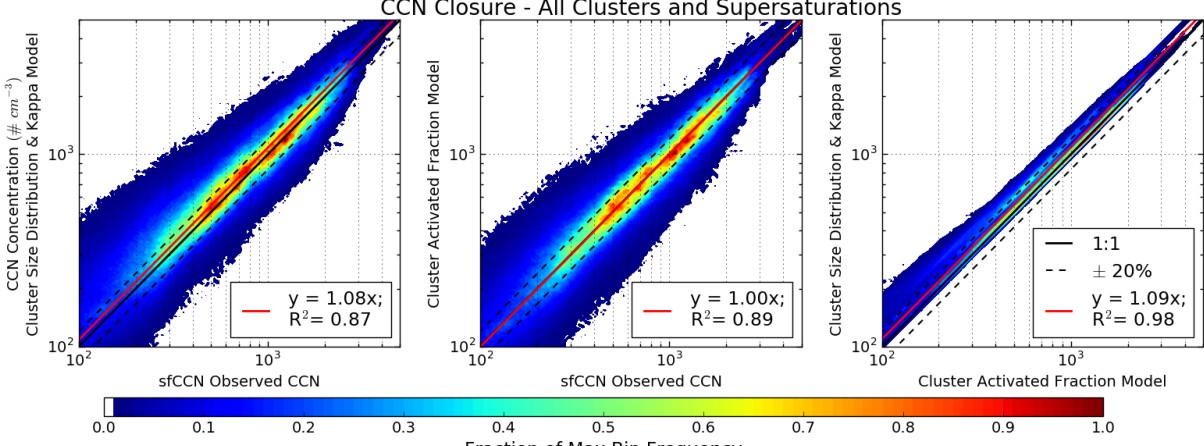

**Figure 7** **CCN closure comparing predicted and measured CCN number concentrations for (a) the predicted concentration based on the size distribution and kappa reconstruction against observed concentrations, (b) the predicted concentration based on the best-fit activated fraction reconstruction against observed concentrations, and (c) the comparison between the two reconstruction methods. All cluster types and supersaturations across the full range of observed values are shown. Linear best-fit slope is shown in red with the associated R-squared values. One to one lines and lines at ± 20% are shown.**



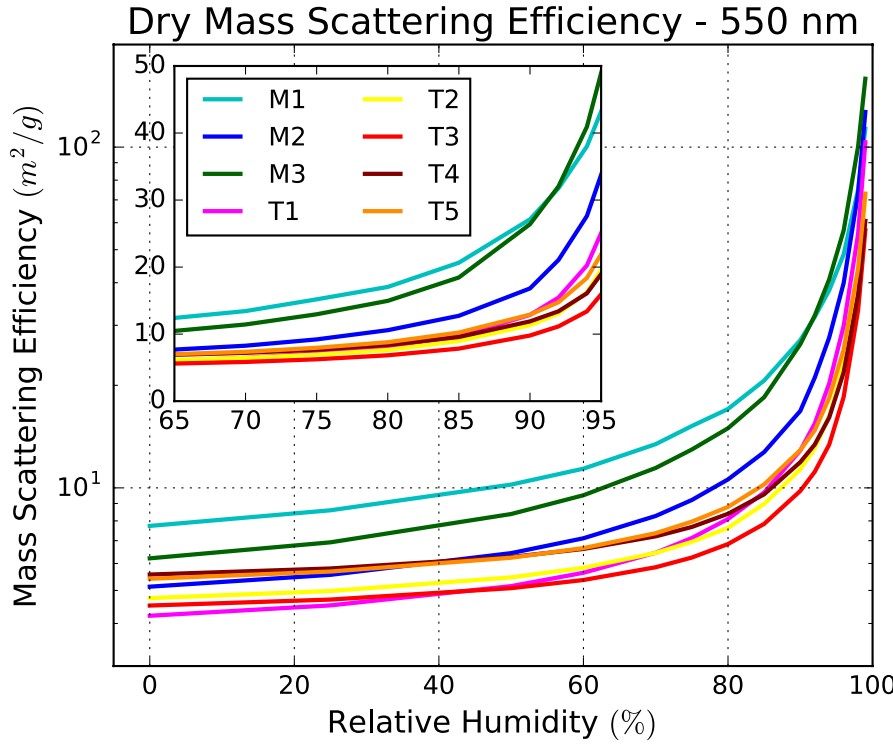

**Figure 8 Reconstructed mass scattering efficiencies for each of the cluster population types across a range of environmental relative**
**humidity values. An assumed dry index of refraction of 1.5 + 0.0i was used for all population types to highlight the differences in**
**aerosol optical properties expected due to differences in population average size distribution, hygroscopicity, and relative humidity.**

