# Peer review of "Flow Scanning CCN (sfCCN) System Calibration"

_Atmospheric Chemistry and Physics, 2018_

## Referee Comment (RC1) · Anonymous Referee #1 · 3 Feb 2019

This is a review of "Classification of aerosol population type and cloud condensation nuclei properties in a coastal California littoral environment using an unsupervised cluster model" by Atwood et al. This manuscript reports on particle size distributions and cloud condensation nuclei measurements made during a field campaign run during CalWater-2015. After cleaning the data the authors identify eight distinct clusters of particle types which they believe are statistically distinct.

The manuscript is clearly deserving of publication and I see no major issues with either the experimental technique or the data reduction and processing techniques. However,

I do see the need to further explain several of their techniques and choices and to clarify some areas.

POINTS OF CLARIFICATION (MAJOR AND MINOR):

Pp1 line 29: The sentence starting on this line needs to be reworded. It is not readily understandable upon a reading.

Pp2 line 4: ACAPEX is not defined upon initial use.

Pp2 line 19: A source should be given for the sentence that starts on this line.

Pp3 line 15, Pp4 line 28: The references in this paper require some attention. In these lines, the punctuation appears incorrect.

Pp4 line 7: Information about the relative smoothness of size distribution of the local contaminant would lend further credibility to this claim. A local contaminant would often have a choppier character in the individual distributions as compared to particles that are more aged and more processed.

Pp6, section starting 2.4.1 requires more attention than the rest of the paper. These sections require significant clarification. -It appears that the authors used a hierarchical clustering scheme to identify the ideal number of clusters. How was this done? Was it bootstrapped? What portion of the data was used? If all the data why then go back to a k-means scheme? -There is inconstancy in the reported number of variables being used in the classification. In line 14 it is reported as 24, in line 24 it is suggested that it is 20. These should be made consistent AND exactly what each of these 20 or 24 variable is should be clearly identified. -The choice of a Pearson Euclidian distance is confusing in this case and should be further justified. Is this the same thing as a Karl Pearson distance in which weights are usually standardized by standard deviation? If normalizing the weights why not just use a Euclidian distance? If a true Karl Pearson Euclidian distance function is being used why aren't the weights a reciprocal of variance rather than evenly distributed by the number of variables? This section requires

ACPD
significant justification and explanation. -In line 20, missing data are referenced. The fraction of missing data and which variables are most often missing should be specified.

In line 8 (Pp7), the "physical interpretability" should be clarified. Is this just empirical judgment? Pp7 line 21 and Pp11 lines 13 and 14: The reference manager again seems to be troubled. These references need to be fixed.

Pp11 line1: The modal kappas should be reported as 0.3-0.5 or 0.30-0.54.

Pp11 line 2: Is there a data associated with Phillips paper?

Pp22 Figure 2: The CN concentration measurements might be better read in a table rather than represented this way. This is especially true in figure c in which they appear to run over the top of the y axis.

Despite these areas that require clarification this paper should clearly still be published after these minor modifications are made and I congratulate the authors on their work.

**ACPD**

---

## Referee Comment (RC2) · Anonymous Referee #2 · 7 Feb 2019

Results of a measurement study during the CalWater-2015 campaign are presented for aerosol size distributions and cloud condensation nuclei. The authors performed a cluster analysis to analyze the data according to air mass type. The identified several aerosol population types that experienced impacts from both marine and terrestrial sources. They also identified populations that were affected by different aging or cloud processing. The paper is useful to the community for identifying differences in aerosol properties associated with terrestrial and marine influences, including different influences within marine air masses associated with varying degrees of cloud processing

and removal by precipitation. The paper is well organized and well written and I recommend its publication after addressing minor comments below.

Page 1/13: Include the season when the study occurred after so the readers can place the rest of the abstract in context.

Page 1/22: Replace "region" with "regions"

Page 1/33: The abstract could be a bit stronger if there was a statement or description of the larger implications or purpose of the study.

Page 2/4: Please spell out "ACAPEX"

Page 2/19: What do the authors mean here by "distribution-averaged" kappa?

Page 2/22-25: This sentence is a little unclear. Wouldn't organic aerosol necessarily be influenced by organic material?

Page 2/28-30: This sentence is along the lines of what would help the abstract be stronger.

Page 2/31: Later in this paragraph a couple of dates are given regarding previous work. It would help to provide the overall dates of the study here to put those into context.

Page 3/1: What kind of data?

Page 3/6: "outside those periods"- see earlier comment on providing the dates of the study earlier, perhaps at the beginning of this paragraph.

Page 3/6: Again, a larger implication statement here would help motivate the work. Perhaps move the lines 28-30 on page 2 to here.

Page 3/13: Is this 5 m above the roof? So total height above ground is?

Page 4/4: I assume these are all number median diameters and number geometric standard deviations?

Page 4/9: I'm not sure what the authors mean by smallest fitted mode's dN/dlogDp. At what size? Do they mean this corresponds to the value of dN/dlogDp at the smallest mode's median diameter?

Page 4/17: Include "contamination" between "local" and "mode"

Page 4/26: Again, what is meant by "distribution-averaged"?

Page 5/7: Refer to figures as Fig. 2a and Fig. 2b.

Page 5/10: It would help to move these references to after the descriptions of the figure, so "Fig. 2c" to line 11 and Figure 2d to line 12.

Page 5/14: Change "Fig. 2(e&f) to "Fig. 2e and Fig 2f., respectively).

Page 6/7: Was this done using the fit lines shown in Figure 2?

Page 6/17: Please include what "i" and "j" refer to.

Page 7/19: Add "number" after "best-fit"

Page 7/22: Please move the figure part (a,b, etc.) before the thing in the figures, so "(a) aerosol and (b) meteorological..."

Page 7/23: Add "number" before "size distributions"

Page 7/31: Is the normalized size distribution the average of lognormals or the all size distributions averaged and then fit?

Page 10/3: This sentence is somewhat unclear- I think the authors mean that "...Fig. 3 indicated that the size distributions associated with the T1 cluster grew...", not that that T1 cluster grew.

Page 11/2: Add "(2018)" after "Phillips et al."

Page 11/4: Same comment as for line 2.

Page 11/30: Add "(2010)" after "Kammermann et al."

Page 12/20: What RH is considered "dry" here?

Page 12/24: This is true, but super-micron particles also are typically associated with lower mass scattering efficiencies.

Page 14/9: Check the website provided for data repository. I assume this will be updated upon final submission.

Page 20/ 22: Table 1 caption: Add "percentage of observations" to the caption list and remove the "Best-fit size distribution. . ." sentence.

Page 20/26: Table 2 caption: Include "number" for "best-fit number size distribution". The caption could be expanded to include something about the cluster types (what 'M' and 'T' mean), different modes and describe "a,b,c" (form of equation)

Page 27: Figure 7: State in the caption what the colors represent.

Page 28: Figure 8: The title of this figure reads "dry", although these are clearly a function of relative humidity? Supplemental

References, Line 36-27; 47: Check reference
* * *

---

## Author Comment (AC1) · 26 Mar 2019

Our response to comments and updates to the manuscript can be found in the attached file.

Please also note the supplement to this comment:
https://www.atmos-chem-phys-discuss.net/acp-2018-1297/acp-2018-1297-AC1-supplement.pdf

---

## Author Response (AR1)

Author Response acp-2018-1297

We thank the two anonymous peer reviewers for their helpful comments and suggestions. Each of the reviewer comments is listed below in bold, followed by our response and the changes made to address each comment in plain text. In some cases, comments which address similar issues were reordered for clarity. In these cases, relevant corrections that were made for multiple comments are listed after each comment.

An updated version of the manuscript with changes highlighted follows the comment responses.

**Anonymous Referee #1 Comments**

**Pp1 line 29: The sentence starting on this line needs to be reworded. It is not readily understandable upon a reading.**

This sentence and the remainder of the abstract were simplified and clarified to read:

"Differences between many of the terrestrial and marine population types in total CCN number concentrations active at a specific supersaturation were often not as pronounced as the associated differences in the corresponding activated fraction spectra, particularly for supersaturations below about 0.4%. This finding was due to the generally higher number concentrations in terrestrial airmasses offsetting the lower fraction of particles activating at low supersaturations. At higher supersaturations, CCN concentrations for aged terrestrial types were typically above those of the marine types due to their higher number concentrations."

**Pp2 line 4: ACAPEX is not defined upon initial use.**

This was updated with the full ACAPEX title as:

"U.S. Department of Energy Atmospheric Radiation Measurement (ARM) Climate Research Facility Cloud-Aerosol-Precipitation Experiment (ACAPEX)"

**Pp2 line 19: A source should be given for the sentence that starts on this line.**

This sentence was updated as:

"As the aerosol ages, chlorine replacement by uptake of acidic gases can reduce the hygroscopicity of the salts (Finlayson-Pitts and Pitts, 1999; Song and Carmichael, 1999)."

**Pp3 line 15, Pp4 line 28: The references in this paper require some attention. In these lines, the punctuation appears incorrect.**

**Pp7 line 21 and Pp11 lines 13 and 14: The reference manager again seems to be troubled. These references need to be fixed.**

The reference manager was evidently creating broken links. Reference fields throughout the manuscript have now been regenerated, updated, and locked to prevent further errors.

**Pp4 line 7: Information about the relative smoothness of size distribution of the local contaminant would lend further credibility to this claim. A local contaminant would often have a choppier character in the individual distributions as compared to particles that are more aged and more processed.**

This is a particularly interesting suggestion for improvements to this methodology, as many of the individual size distributions may indeed have what could be identified as essentially increased noise in the measured values. However, in some cases of influence from a local contaminant an increased signal to noise ratio or similar metric did not occur. For instance, in the example number size distribution spectra from Fig. 1 there was some noise or lack of smoothness in the measured values at the smallest bin sizes for the non-contaminated spectrum (Fig. 1a), while a relatively smooth spectrum was seen in the small contamination mode a short time later (Fig. 1b). This was in part due to higher levels of noise in the smallest bin sizes measured by the SMPS (e.g. the individual distributions in Fig. 4), particularly when few particles were observed in these bins. The effect of expected increased noise associated with fresh contaminant modes was therefore competing with the effect of increased measurement noise at small bin sizes in some of the SMPS data points. As a result, a reliable criterion associated with this behavior could not be established in this dataset and we therefore opted not to try to incorporate such a metric into our methodology in this work.

The following text was added to the supplemental materials section on removal of contamination modes to address this consideration:

"Further criteria for identification of contamination periods and modes in the number size distribution were considered, including analysis of signal-to-noise or similar measures of relative smoothness of potential contamination modes. However, such metrics were not always consistent in their identification of such small contamination modes due to competing factors from measurement noise and physical variability. Ultimately, the criteria listed in the main text were found to be sufficiently capable of removing small contamination modes such that the cluster model was able utilize these data points without skewing the results, and so were used for the purposes of this analysis."

**Pp6, section starting 2.4.1 requires more attention than the rest of the paper. These sections require significant clarification.**

**-It appears that the authors used a hierarchical clustering scheme to identify the ideal number of clusters. How was this done? Was it bootstrapped? What portion of the data was used? If all the data why then go back to a k-means scheme?**

The purpose of the hierarchical clustering scheme was to help identify the appropriate number of clusters to use for the K-means clustering, and was not otherwise related to the K-means clustering methodology. An agglomerative clustering method was used for this hierarchical clustering using the same distance function as the K-means analysis, wherein subsequent steps merged the two data points or clusters with the smallest value of the distance function between them until only one cluster remained. A dendrogram was then used to check for increases in distance between subsequent clustering steps. Steps with the largest increase were treated as potential candidate numbers of cluster for the subsequent K-means analysis. This process for selecting potential numbers of clusters appropriate for the K-means analysis was clarified at the end of second 2.4 as:

"Selection of the appropriate number of clusters for the K-Means analysis followed a similar methodology to that described in Atwood et al., (2017), and was based on both an initial hierarchical clustering and the use of internal validity measures (Wilks, 2011). First, a hierarchical agglomerative cluster analysis was created using the cluster.AgglomerativeClustering class of scikit-learn to merge the two data points or clusters with the smallest value of the distance function together into a single cluster in subsequent steps until only one cluster remained. A dendrogram was then used to identify potential numbers of clusters at agglomeration steps that had the largest increase in distance between merged clusters. Various internal validity measures of clusters in the ensuing K-Means clustering (Beddows et al., 2009; Baarsch and Celebi, 2012) were then used to further assess appropriate numbers of clusters. The final selection of appropriate cluster numbers was based on each of these measures, along with verification that the results maintained physically distinct and temporally coherent clusters, in order to select the appropriate number of K-means clusters."

**-There is inconstancy in the reported number of variables being used in the classification. In line 14 it is reported as 24, in line 24 it is suggested that it is 20. These should be made consistent AND exactly what each of these 20 or 24 variable is should be clearly identified.**

These various values are the number of variables in several categories of variables used in the clustering. These included 20 variables for the size distribution, 20 variables for the activated fraction distribution, 24 variables for the HYSPLIT trajectories, 2 variables for the local wind speed, and 1 variable for the total number concentration, for a total 67 clustering variables used for each data point in the distance function.

This was clarified in section 2.4.1 and now reads:

"A total of 67 clustering variables were used to describe each time stamp in this analysis, which included measurements of aerosol microphysical properties and meteorological parameters at BML. Aerosol property variables included the normalized size distribution (normalized to an integrated value of 1 cm$^{-3}$ by dividing by the total particle number concentration), after correction for local contamination. Values of the normalized number size distributions (dN/dlog$_{10}$Dp) were discretized into 20 logarithmically spaced bins, with each mean bin value then serving as a separate variable in the clustering distance function (e.g. Charron et al. (2008)). Similarly, activated fraction spectra (the fraction of particles activated at supersaturation S) for each time stamp were divided into 20 linearly spaced bins distributed between 0% and 1.1% supersaturation to incorporate CCN properties into the analysis. Total particle number concentration was included as a separate cluster variable.

Meteorological parameters at BML included the local 10m observed wind velocity as perpendicular u and v component variables (2 variables). Additionally, HYSPLIT backtrajectories were assigned to data points closest in time to each trajectory. The backtrajectory was converted to separate variables for the distance function by determining the distance from the receptor, initial bearing, and altitude, every three hours backwards along the trajectory for 24 hours, yielding a total of 24 trajectory clustering variables for each time stamp."

**-The choice of a Pearson Euclidian distance is confusing in this case and should be further justified. Is this the same thing as a Karl Pearson distance in which weights are usually standardized by standard deviation? If normalizing the weights why not just use a Euclidian distance? If a true Karl Pearson Euclidian distance function is being used why aren't the weights a reciprocal of variance rather than evenly distributed by the number of variables? This section requires significant justification and explanation.**

We agree that this terminology was confusing and in need of correction, and appreciate the efforts of the reviewer in identifying this. In particular, we did in fact first standardize each of the variables using the standard deviation to account for the different scales of the clustering variables. As such and as noted by the reviewer, we should have referred to this distance function as the "Karl Pearson distance" rather than the "Karl Pearson Euclidian distance". This has been corrected.

We further weighted each of the variables based on the variable categories noted in the previous section. As there was a total of 67 variables, but only five categories of variables (size distribution, activation spectra, number concentration, wind speed, and backtrajectory), the purpose of this additional weighting was to reduce the impact of differing numbers of variables within categories, similar to the intent of the Karl Pearson standardization.

This section has been changed to explain the method more clearly:

"As variables of different scale were used in the cluster modelling, the Karl Pearson distance function (Wilks, 2011) was used, wherein each variable is first standardized to ensure they have equal weight on the distance function. However, as the 67 clustering variables were grouped into five categories of measurements (number size distribution, activation spectrum, total particle number concentration, wind speed, and backtrajectory), a further modification to the weighting (described below) was used to reduce the impact of differing numbers of variables in these categories on the distance function. The resulting distance between any two data points i and j was given by the function:

$$d_{i,j} = \left[ \sum_{k=1}^{K} w_k \left( x_{i,k} - x_{j,k} \right)^2 \right]^{\frac{1}{2}} \tag{1}$$

where $d_{(i,j)}$ gives the distance between two data point vectors, $x\_i$ and $x\_j$, in a K-dimensional space (i.e. K nominally independently measured or observed, orthogonal variables at each time stamp), for each variable, k, with weight, $w\_k$, calculated for each variable by:

$$w_k = \frac{v_k}{s_k} \tag{2}$$

where $s\_k$ is the variance of the variable k, and $v\_k$ is the further relative weight used for each variable.

Only data points with valid measurements for number size distribution and activated fraction were used in the clustering analysis, leaving approximately 94% of data points (3357 of a total of

3583) utilized. Of the remaining data, two data points had partially invalid activation spectra and were kept in the analysis. The partial spectra had invalid data points imputed to the variable mean to minimize their impact on the distance function.

The size distribution and activated fraction variable categories, each of which had 20 variables, were given relative weights, $v\_k$, of 1/20 such that their total relative weights summed to 1. As these variables were of primary importance to aerosol population microphysical properties, the other variable groups were decreased in relative importance. The backtrajectory and wind vector categories were each assigned a relative weight of 0.5, and the total particle number concentration variable assigned a relative weight of 0.1. As cluster analysis is by its nature an exploratory data analysis technique, these relative weight values were reached by varying their values and assessing the physical interpretability of the results (discussed further in the next section)."

**-In line 20, missing data are referenced. The fraction of missing data and which variables are most often missing should be specified.**

In this analysis, only data points with valid number size distribution and activation spectra data were utilized in the cluster analysis, leaving approximately 94% of data points (3357 of 3583). Of these data points, two had only partially valid activation spectra. Invalid values for these spectra were imputed to the variable mean value for the purposes of the distance function. All other clustering variables had valid data for each clustering data point. This information was added to the text as:

"Only data points with valid measurements for number size distribution and activated fraction were used in the clustering analysis, leaving approximately 94% of data points (3357 of a total of 3583) utilized. Of the remaining data, two data points had partially invalid activation spectra and were kept in the analysis. The partial spectra had invalid data points imputed to the variable mean to minimize their impact on the distance function."

**In line 8 (Pp7), the "physical interpretability" should be clarified. Is this just empirical judgment?**

The main criteria for assessing the proper number of clusters in this analysis were temporal coherence of the clusters (i.e. data points assigned to specific clusters occurring sequentially or close in time, rather than randomly distributed throughout the study period), and clusters with physically interpretable or meaningful results (i.e. the cluster results could be related to reasonable physical interpretations such as changes in cluster types associated with land/sea-breeze shifts).

The meaning of "physically meaningful" has been clarified and expanded upon by adding the following to the first description of these criteria in this section as:

"… temporally coherent (i.e. data points assigned to specific clusters tended to occur near each other rather than randomly distributed throughout the study period) and physically meaningful (i.e. could be related to physical phenomena such as land/sea-breeze shifts) …"

In addition, the last sentence of this section with the noted reference to "physical interpretability" has been clarified as follows:

"The eight-cluster option had been initially identified as potentially appropriate based on the internal validity measures and hierarchical clustering, while other numbers of clusters did not improve results based on the criteria of temporally coherent clusters and physically meaningful interpretation of the cluster results. As such, the eight-cluster model was selected as the most appropriate unsupervised classification result."

**Pp11 line1: The modal kappas should be reported as 0.3-0.5 or 0.30-0.54.**

We have made this correction.

**Pp11 line 2: Is there a data associated with Phillips paper?**

The comparison to the Phillips et. al., (2018) results was corrected and clarified as follows:

"Mean measured $\kappa$ values of 0.49 and 0.46 for the M1 and M3 population types, respectively, were within the range of mean $\kappa$ values (0.30 to 0.54) for marine aerosol dominated periods in a coastal outflow marine region presented by Phillips et al., (2018). Mean values for terrestrial population types T1–T5 varied between 0.15 and 0.25, and were consistent with the 0.20 value for Aitken mode continental outflow aerosol reported by Phillips et al. (2018)."

**Pp22 Figure 2: The CN concentration measurements might be better read in a table rather than represented this way. This is especially true in figure c in which they appear to run over the top of the y axis.**

We agree that the figure needed to be updated to improve the interpretability of CN concentration. As CN concentration is independent of supersaturation, the point markers have been replaced with a bar in the background highlighting the range of CN observations, and we have included this range as a text value on the plot.

**Anonymous Referee #2 Comments**

**Page 1/13: Include the season when the study occurred after so the readers can place the rest of the abstract in context.**

This has been updated to:

"… are presented for approximately six weeks of observations during the boreal winter/spring as part of the CalWater-2015 field campaign."

**Page 1/22: Replace "region" with "regions"**

This has been corrected.

**Page 1/33: The abstract could be a bit stronger if there was a statement or description of the larger implications or purpose of the study.**

**Page 2/28-30: This sentence is along the lines of what would help the abstract be stronger.**

**Page 3/6: Again, a larger implication statement here would help motivate the work. Perhaps move the lines 28-30 on page 2 to here.**

We appreciate the reviewer's suggestion and have modified the abstract accordingly.

The beginning of the abstract has now been updated to read:

"Aerosol particle and cloud condensation nuclei (CCN) measurements from a littoral location on the northern coast of California at Bodega Bay Marine Laboratory (BML) are presented for approximately six weeks of observations during the boreal winter/spring as part of the CalWater-2015 field campaign. The nature and variability of surface (marine boundary layer, MBL) aerosol populations were evaluated by classifying observations into periods of similar aerosol and meteorological characteristics using an unsupervised cluster model to derive distinct littoral aerosol population types and link them to source regions. Such classifications support efforts to understand the impact of changing aerosol properties on precipitation and cloud development in the region, including during important atmospheric river (AR) tropical moisture advection events."

**Page 2/4: Please spell out "ACAPEX"**

This was updated with the full ACAPEX title as:

"U.S. Department of Energy Atmospheric Radiation Measurement (ARM) Climate Research Facility Cloud-Aerosol-Precipitation Experiment (ACAPEX)"

**Page 2/19: What do the authors mean here by "distribution-averaged" kappa?**

**Page 4/26: Again, what is meant by "distribution-averaged"?**

We agree that this phrasing was confusing and unnecessary, and it has been removed. To clarify: the average value of kappa was obtained by averaging all measurements of hygroscopicity at various supersaturations during the time periods associated with each cluster.

**Page 2/22-25: This sentence is a little unclear. Wouldn't organic aerosol necessarily be influenced by organic material?**

This sentence has been clarified to read:

"In contrast, average hygroscopicities for continental aerosol populations are often in the range of 0.1 to 0.3 due to heavy influence of organic and insoluble material, although individual inorganic sulfates and nitrate salts also found in continental aerosols have higher values, in the same range as many remote marine aerosol populations (Andreae and Rosenfeld, 2008)."

**Page 2/31: Later in this paragraph a couple of dates are given regarding previous work. It would help to provide the overall dates of the study here to put those into context.**

**Page 3/6: "outside those periods"- see earlier comment on providing the dates of the study earlier, perhaps at the beginning of this paragraph.**

We have added the dates used for observations as part of this study and reworded the first sentence in this paragraph to read:

"In this study, observations from 23 January 2015–5 March 2015 at the BML ground site were used to evaluate the nature and variability of surface (marine boundary layer, MBL) aerosol populations."

**Page 3/1: What kind of data?**

The referenced study included other sources of measurements, including other aircraft and ship observations, as part of both the CalWater-2015 and ACAPEX experiments. This has been clarified in this sentence as:

"In prior work analyzing ground-, ship-, and aircraft-based meteorological and microphysical observations from the CalWater-2015 and ACAPEX field experiments, Leung (2016) discusses an AR event that was first observed and sampled from aircraft platforms off the coast of California on 5 February 2015, during the observational period analyzed here."

**Page 3/13: Is this 5 m above the roof? So total height above ground is?**

This height was clarified as:

"… to a height of approximately 5 m above the ground, …"

**Page 4/4: I assume these are all number median diameters and number geometric standard deviations?**
**Page 7/19: Add "number" after "best-fit"**
**Page 7/23: Add "number" before "size distributions"**

This correction has been added to each of these instances to clarify that we are utilizing number size distributions for these parameterizations.

**Page 4/9: I'm not sure what the authors mean by smallest fitted mode's dN/dlogDp. At what size? Do they mean this corresponds to the value of dN/dlogDp at the smallest mode's median diameter?**

This criterion was to check that the distribution fit for the smallest mode constituted less than half to total value at 50 nm. This has been reworded to read:

"The smallest fitted mode constituted less than 50% of the total fitted number concentration at 50 nm (indicating only a small contribution to total particle count at larger sizes)"

**Page 4/17: Include "contamination" between "local" and "mode"**

This has been added.

**Page 5/7: Refer to figures as Fig. 2a and Fig. 2b.**

We have updated this sentence to include this as:

"Example sfCCN concentrations (Fig. 2a) and activated fraction spectra (Fig. 2b) are shown for the same time stamp as the size distribution data in Fig. 1."

**Page 5/10: It would help to move these references to after the descriptions of the figure, so "Fig. 2c" to line 11 and Figure 2d to line 12.**

This change was made to the sentence as:

"During the contamination period approximately one hour later the effect of the small mode noted in Fig. 1b was seen via increased CN concentrations (reaching as high as 5000 cm-3; Fig. 2c) and decreased activated fractions above approximately 0.1% supersaturation (Fig. 2d)."

**Page 5/14: Change "Fig. 2(e&f) to "Fig. 2e and Fig 2f., respectively).**

This change was made as:

"After removal of this small contamination mode the corrected CN concentrations and activation spectrum Fig. 2e and Fig. 2f, respectively, were similar in characteristics to the earlier non-contaminated period."

**Page 6/7: Was this done using the fit lines shown in Figure 2?**

In this sentence the distribution was intended to refer to all data points, which were discretized into bins and then averaged, but we agree that this was confusingly worded. We have updated the sentence to read:

"Values of the normalized number size distributions (dN/dlog$_{10}$Dp) were discretized into 20 logarithmically spaced bins, with each mean bin value then serving as separate variables in the clustering distance function (e.g. Charron et al. (2008))."

**Page 6/17: Please include what "i" and "j" refer to.**

This has been updated to describe the equation as:

"The resulting distance between any two data points i and j was given by the function: …"

**Page 7/22: Please move the figure part (a,b, etc.) before the thing in the figures, so "(a) aerosol and (b) meteorological. . ."**

This has been corrected as:

"Figure 3 presents the study timelines of measured (a) aerosol, and (b) meteorological variables, and the (c) corrected and (d) normalized aerosol size distributions."

**Page 7/31: Is the normalized size distribution the average of lognormals or the all size distributions averaged and then fit?**

All sized distributions were averaged and then fit. This has been clarified earlier in the section as:

"Multi-modal lognormal number size distributions were fit to the average of all spectra associated with each cluster. Fit parameters and best-fit CCN-spectrum activated fraction parameters (see Supplemental Material) for each cluster are provided in Table 2."

**Page 10/3: This sentence is somewhat unclear- I think the authors mean that ". . .Fig. 3 indicated that the size distributions associated with the T1 cluster grew. . .", not that that T1 cluster grew.**

This is correct and we have included this important clarification as:

"During these times, normalized size distributions and modal median diameter fits shown in Fig. 3 indicated that the size distributions associated with the T1 cluster grew over the course of several days into distributions with larger median diameters, which were then classified as other terrestrial cluster types."

**Page 11/2: Add "(2018)" after "Phillips et al."**

**Page 11/4: Same comment as for line 2.**

**Page 11/30: Add "(2010)" after "Kammermann et al."**

The year has been added to correct each of these citations.

**Page 12/20: What RH is considered "dry" here?**

In this analysis the dry values indicate mass scattering efficiencies at 0% RH, with the change in scattering efficiency due to RH reflected by the curves in Fig. 8 (constructed assuming a deliquesced aerosol). In order to better represent this effect, we have made changes to our optical reconstruction and discussion to better reflect the effect of RH on measurements.

The following sentence was added:

"The average size distribution for each aerosol population type was assumed to be made for a deliquesced aerosol at the study average SMPS measured RH of 35.5% (Martin et al., 2017) and the cluster average κ given in Table 1"

In addition, the discussion of dry RH was updated to read:

"Mass scattering efficiencies at 550 nm for particles modeled at 0% RH ranged between 3.6 and 7.6 $m^2$ $g^{-1}$, although actual values would be expected to be lower due to expected particle dry densities higher than 1 g $cm^{-3}$."

The MODIS comparison was also removed as it is not as directly comparable to mass scattering efficiency on a dry aerosol mass basis and could lead to confusion in its interpretation.

**Page 12/24: This is true, but super-micron particles also are typically associated with lower mass scattering efficiencies.**

We agree that this is an important point to make and have updated this to read:

"Further, these values represent only the contribution to mass scattering efficiencies from fine mode aerosol. Coarse mode aerosol, including particles generated by sea spray in littoral environments, can represent a large fraction of the total light scattering, although their mass scattering efficiencies are typically lower."

**Page 14/9: Check the website provided for data repository. I assume this will be up- dated upon final submission.**

This dataset has now been submitted to the repository at doi:10.5281/zenodo.2605668.

**Page 20/ 22: Table 1 caption: Add "percentage of observations" to the caption list and remove the "Best-fit size distribution. . ." sentence.**

This has been corrected and now reads:

"Table 1  Aerosol and meteorological parameters for each of the cluster time periods with the total number and percentage of observations in each cluster. Cluster mean values are given for total particle number concentration, κ hygroscopicity parameter from the srCCN system, HYSPLIT 24-hour accumulated precipitation along the trajectory, and local wind velocity observations."

**Page 20/26: Table 2 caption: Include "number" for "best-fit number size distribution". The caption could be expanded to include something about the cluster types (what 'M' and 'T' mean), different modes and describe "a,b,c" (form of equation)**

These changes have been made and the caption now reads:

"Table 2  Cluster best-fit number size distribution and activated fraction parameters are shown, with activated fraction parameters a, b, and c pertaining to the fit model equation given in the supplemental information. Clusters with "M" and "T" names refer to marine and terrestrial aerosol population types, respectively."

**Page 27: Figure 7: State in the caption what the colors represent.**

The following sentence was added to the caption:

"Colors represent the data point density as a fraction of the maximum density in each plot."

**Page 28: Figure 8: The title of this figure reads "dry", although these are clearly a function of relative humidity?**

The "dry" was intended to indicate the mass scattering efficiency was calculated on a dry aerosol mass basis, but we agree it is confusing. Instead we have changed the figure to read "Scattering efficiency per unit dry mass– 550 nm", and indicated that this value is calculated against total dry particle mass in the caption as:

"Figure 8  Reconstructed mass scattering efficiencies (per unit dry aerosol mass) for each of the cluster population types across a range of environmental relative humidity values. An assumed dry index of refraction of $1.5 + 0.0i$ was used for all population types to highlight the differences in aerosol optical properties expected due to differences in population average size distribution, hygroscopicity, and relative humidity."

**Supplemental References, Line 36-27; 47: Check reference**

The reference manager was evidently creating broken links. Reference fields throughout the manuscript have now been regenerated, updated, and locked to prevent further errors.

**Classification of aerosol population type and cloud condensation nuclei properties in a coastal California littoral environment using an unsupervised cluster model**

Samuel A. Atwood[1], Sonia M. Kreidenweis[1], Paul J. DeMott[1], Markus D. Petters[2], Gavin C. Cornwell[3], Andrew C. Martin[4], Kathryn A. Moore[1,3]

[1]Department of Atmospheric Science, Colorado State University, Fort Collins, CO 80523, USA
[2]Department of Marine, Earth and Atmospheric Sciences, North Carolina State University, Raleigh, NC 27695, USA
[3]Department of Chemistry and Biochemistry, University of California San Diego, La Jolla, CA, USA
[4]Climate Atmospheric Science and Physical Oceanography, Scripps Institution of Oceanography, La Jolla, CA, USA

*Correspondence to*: Sonia M. Kreidenweis (soniak@colostate.edu)

**Abstract.** Aerosol particle and cloud condensation nuclei (CCN) measurements from a littoral location on the northern coast of California at Bodega Bay Marine Laboratory (BML) are presented for approximately six weeks of observations during the  boreal winter/spring as part of the CalWater-2015 field campaign. The nature and variability of surface (marine boundary layer, MBL) aerosol populations were evaluated by classifying observations into periods of similar aerosol and meteorological characteristics using an unsupervised cluster model to derive distinct littoral aerosol population types and link them to source regions. Such classifications support efforts to understand the impact of changing aerosol properties on precipitation and cloud development in the region, including during important atmospheric river (AR) tropical moisture advection events. Eight aerosol population types were identified that were associated with a range of impacts from both marine and terrestrial sources. Average measured total particle number concentrations, size distributions, hygroscopicities, and activated fraction spectra between 0.08% and 1.1% supersaturation are given for each of the identified aerosol population types, along with meteorological observations and transport pathways during time periods associated with each type. Five terrestrially influenced aerosol population types represented different degrees of aging of the continental outflow from the coast and interior of California, and their appearance at the BML site was often linked to changes in wind direction and transport pathway. In particular, distinct aerosol populations, associated with diurnal variations in source  regions induced by land/sea-breeze shifts, were classified by the clustering technique. A terrestrial type representing fresh emissions, and/or a recent new particle formation event, occurred in approximately 10% of the observations. Over the entire study period, three marine influenced population types were identified that typically occurred when the regular diurnal land/sea-breeze cycle collapsed and BML was continuously ventilated by air masses from marine regions for multiple days. These marine types differed from each other primarily in the degree of cloud processing evident in the size distributions, and in the presence of an additional large-particle mode for the type associated with the highest wind speeds. One of the marine types was associated with a multi-day period during which an atmospheric river made landfall at

BML. Differences between
many of the terrestrial  types  in total CCN number concentrations
active at a specific supersaturation were often not as pronounced as the associated differences in the corresponding
activated fraction spectra, particularly for supersaturations below about 0.4%.
This finding was due to the generally higher number concentrations in terrestrial
airmasses offsetting the lower fraction of particles activating at low supersaturations. At higher supersaturations, CCN
concentrations for aged terrestrial types were typically above those
of the marine types due to their higher number concentrations .

**1    Introduction**

Atmospheric rivers (ARs) are tropical moisture advection phenomena that can account for large fractions of the wintertime precipitation in California (Ralph et al., 2004; Dettinger et al., 2011). The winter-spring 2015 CalWater-2015 study (Ralph et al., 2015), and coordinated U.S. Department of Energy Atmospheric Radiation Measurement (ARM) Climate Research Facility Cloud-Aerosol-Precipitation Experiment (ACAPEX) (
[revised manuscript text omitted]
, and was based on both an initial hierarchical clustering and the use of internal validity measures (Wilks, 2011). First, a hierarchical agglomerative cluster analysis was created using the *cluster.AgglomerativeClustering* class of scikit-learn to merge the two data points or clusters with the smallest value of the distance function together into a single cluster in subsequent steps until only one cluster remained. A dendrogram was then used to identify potential numbers of clusters at agglomeration steps that had the largest increase in distance between merged clusters. Various internal validity measures of clusters in the ensuing K-Means clustering (Beddows et al., 2009; Baarsch and Celebi, 2012) were then used to further assess appropriate numbers of clusters. The final selection of appropriate cluster numbers was based on each of these measures, along with verification that the results maintained physically distinct and temporally coherent clusters, in order to select the appropriate number of K-means clusters.

**2.4.1 Cluster variables**

VariablesA total of 67 clustering variables were used to describe each time stamp in this analysis, which included measurements of aerosol microphysical properties and of meteorological parameters at BML. Aerosol property variables included the normalized size distribution at each time stamp (normalized to an integrated value of 1 cm$^{-3}$ by dividing by the total particle number concentration), after correction for local contamination. TheValues of the normalized number size distributions (dN/dlog$_{10}$Dp) were discretized into 20 logarithmically spaced bins that served, with each mean bin value then serving as a separate variablesvariable in the clustering distance function (e.g. Charron et al. (., 2008)). ActivatedSimilarly, activated fraction spectra (the fraction of particles activated at supersaturation S) for each time stamp were divided into 20

linearly spaced bins distributed between 0% and 1.1% supersaturation to incorporate CCN properties into the analysis. Total particle number concentration was included as a separate cluster variable.

Meteorological parameters at BML included the local 10m observed wind velocity as perpendicular u and v component variables. (2 variables). Additionally, HYSPLIT backtrajectories were assigned to data points closest in time to each trajectory.

The backtrajectory was converted to separate variables for the distance function by determining the distance from the receptor, initial bearing, and altitude, every three hours backwards along the trajectory for 24 hours, yielding a total of 24 trajectory clustering variables for each time stamp.

**2.4.2   Distance function**

TheAs variables of different scale were used in the cluster modelling, the Karl Pearson Euclidean distance function (Wilks,

2011) used in the cluster model was modified to include a relative weight parameter forwas used, wherein each input variable is first standardized to ensure they have equal weight on the distance function. However, as the 67 clustering variables were grouped into five categories of measurements (number size distribution, activation spectrum, total particle number concentration, wind speed, and backtrajectory), a further modification to the weighting (described below) was used to reduce the impact of differing numbers of variables in these categories on the distance function. The resulting distance between any two data points i and j was given by the function:

$$d_{i,j} = \left[ \sum_{k=1}^{K} w_k \left( x_{i,k} - x_{j,k} \right)^2 \right]^{\frac{1}{2}} \tag{1}$$

where $d_{i,j}$ gives the Euclidean distance between two data point vectors, $x_i$ and $x_j$, in a $K$-dimensional space (i.e. $K$ nominally independently measured or observed, orthogonal variables at each time stamp), for each variable, $k$, with relative weight, $w_k$.

Each, calculated for each variable was first standardized, with missing data imputed to values of zero to minimize their impact on the distance function.by:

The

$$w_k = \frac{v_k}{s_k} \tag{2}$$

where $s_k$ is the variance of the variable k, and $v_k$ is the further relative weightsweight used for each variable.

Only data points with valid measurements for number size distribution and activated fraction were included to prevent properties of the aerosol from becoming over-weightedused in the cluster model dueclustering analysis, leaving approximately

94% of data points (3357 of a total of 3583) utilized. Of the remaining data, two data points had partially invalid activation spectra and were kept in the analysis. The partial spectra had invalid data points imputed to the variable mean to having more variables describing them. minimize their impact on the distance function.

The size distribution and activated fraction variable categories, each of which had 20 variables, were given relative weights, $v_{k}$, of 1/20 such that their total relative weights summed to 1. As these variables were of primary importance to aerosol population microphysical properties, the other variable groups were decreased in relative importance. The backtrajectory and wind vector categories were each assigned a relative weight of 0.5, and the total particle number concentration variable assigned a relative weight of 0.1. As cluster analysis is by its nature an exploratory data analysis technique, these relative weight values were reached by varying their values and assessing the physical interpretability of the results (discussed further in the next section).

**2.4.3    Number of clusters**

Hierarchical clustering and internal validity measures indicated two, six, eight, and twelve clusters were potentially appropriate for the K-means analysis. Clusters associated with periods of marine aerosol impacts (discussed further in the next section)

became temporally coherent (i.e. data points assigned to specific clusters tended to occur near each other rather than randomly distributed throughout the study period) and physically meaningful (i.e. could be related to physical phenomena such as land/sea-breeze shifts) after the number of clusters was increased to eight.

In the case of the twelve-cluster analysis, several of the clusters were composed of few or even a single data point, indicating the model had begun to separate outliers into distinct clusters. In addition, several of the temporally consistent clusters were split, indicating that too many clusters had been selected. All potential cluster numbers between seven and eleven were then investigated to determine if physically or temporally coherent population types emerged to a greater degree than the eight- cluster model. The eight-cluster option had been initially identified as potentially appropriate based on the internal validity measures and  hierarchical clustering, while other numbers of clusters did not improve results based on the criteria of temporally coherent clusters and physically meaningful interpretation of the cluster results As such, the eight-cluster model was selected as the most appropriate unsupervised classification result.

**3    Aerosol population type classification results and discussion**

Three of the eight identified clusters were defined as "marine" population types, as backtrajectory data showed evidence of transport pathways primarily over ocean areas. These marine types, denoted as clusters M1–M3, tended to have lower average number concentrations (below approximately 1500 cm$^{-3}$), while the terrestrial clusters (T1–T5) had typical averages between approximately 2000 and 4000 cm$^{-3}$ and were associated with transport from more terrestrial source regions. The exception to this was cluster T5, which had number concentrations of roughly 1500 cm$^{-3}$, more oceanic transport pathways, and size distributions with the largest median diameters among the "terrestrial" clusters. Table 1 provides the cluster-averaged number concentrations, wind velocities, HYSPLIT accumulated precipitation along the 24-hour trajectory, hygroscopicity parameters from the srCCN system, and the percentage of all measurements associated with each of the 8 identified clusters.

Multi-modal lognormal number size distributions were fit to the average of all spectra associated with each cluster.

[revised manuscript text omitted]

. Mean values for terrestrial population types T1–T5 varied between 0.15 and 0.25, and were consistent with the 0.20 value for Aitken mode continental outflow aerosol reported by Phillips et al. (2018). However, we note that $\kappa$ values measured in this study and used in the closure calculations were derived from supersaturated CCN

measurements, whereas Phillips et al. (2018) reported $\kappa$ from humidified growth factors at 80% RH. Prior work has shown that $\kappa$ is not always consistent across this large of a span of RH (Irwin et al.,

2010; Whitehead et al., 2014), thus contributing to uncertainty in such comparisons. The hygroscopicity for the final marine

M2 type was 0.30, near the lower end of typical values for marine aerosol in regions of continental outflow, but still above those of the terrestrial population types. As the M2 cluster was also the only marine population type with no indication of recent precipitation scrubbing of the airmass prior to arrival at BML (Figure 3b), some combination of influences from cloud processing, marine, and terrestrial or anthropogenic sources may result in the observed hygroscopicity values between those of the other population types.

The cluster-average hygroscopicities from Table 1 were combined with the average cluster size distributions from Table 2 to create a reconstructed activated fraction spectrum for each cluster. These reconstructions are compared with the direct-

1. measured CCN spectra in Fig. S3 and Table 2. Generally, the reconstructed spectra are within one standard deviation of the

2. directly measured spectra from the sfCCN system. However, the reconstruction overpredicts activated fraction for the marine

3. clusters, with the largest discrepancies at low supersaturations and low CCN number concentrations. Similar behaviors in

4. overprediction of CCN concentrations based on reconstructions using hygroscopicity and size distributions have been noted

5. before (McFiggans et al., 2006). Kammermann et al., (2010) reviewed a number of studies that compared such predicted CCN

6. concentrations against observed values and found biases were often largest at supersaturations below 0.3%, where predictions

7. deviated from observations by factors ranging from 0.6 to 3.3, though with most studies finding overprediction occurred. They

8. attributed the increasing bias at decreasing supersaturations to increased uncertainty in the critical activation diameter and

9. associated CCN number prediction, though they also noted that this discrepancy between predicted and observed CCN

10. concentrations has not been fully resolved. At larger particle diameters measurement uncertainty increases due to imprecise

11. particle size cuts, losses in inlets and tubing, and inversion uncertainties, along with generally lower number concentrations

12. than at smaller particle diameters, leading to higher expected uncertainty in CCN predictions and reconstructions when the

13. critical activation diameter is in this range of particle sizes. At low supersaturations where BML data showed an overprediction

14. bias the critical activation diameter would be above 150 nm. The marine types that had the largest biases at low supersaturations

15. also tended to have larger fractions of particles at these large sizes. This would be expected to add to uncertainty in ways

16. similar to those noted by Kammermann et al.,

17. (2010) and potentially explain the larger discrepancies in CCN prediction at lower

18. supersaturations in the marine aerosol types.

19. Further investigation of the closure between predicted and observed CCN concentration was conducted using two prediction

20. models. Hygroscopicity derived predictions using cluster average $\kappa$ and normalized size distribution values were generated

21. and compared against all observed CCN concentrations by the sfCCN system in Figure 7(a). Similarly, the predicted CCN

22. concentration using cluster average activated fraction spectra were compared against observations in Figure 7(b). Both models

23. predicted the activated fraction using the cluster type identified during the observation, which was then multiplied by the

24. observed total number concentration at the observation time. The hygroscopicity and size model showed overprediction

25. compared to both observations and the activated fraction model (Figure 7c), with best-fit slopes of 1.08 and 1.09 respectively.

26. The activated fraction model predictions did improve on the hygroscopicity model, with a slope of 1.00 and $R^2$ values

27. increasing from 0.87 to 0.89, though this is in part due to the model being based on a direct fit of the observed data.

28. Nevertheless, the degree of closure between model predictions and observed CCN concentrations is similar to closures

29. previously reported in field studies (e.g. Bougiatioti et al., 2009; Kammermann et al., 2010).

30. ### 3.5    Aerosol optical properties

31. While optical properties of the various aerosol population types were not directly measured, a simple optical reconstruction

32. was conducted to evaluate potential differences between the population types due to differences in size distribution and

33. hygroscopicity. Average particle size distributions and measured average $\kappa$ values were used to grow particles to equilibrium with relative humidity across a range of values between 0% and 99%, assuming deliquesced particles, followed by estimation of mass scattering efficiency on a dry aerosol mass basis using Mie theory (Bohren and Huffman, 1983). For the purposes of this simple optical comparison, supersaturated $\kappa$ values used for this analysis were assumed to be sufficient to provide an estimate of subsaturated hygroscopic growth. The average size distribution for each aerosol population type was assumed to be made for a deliquesced aerosol at the study average SMPS measured RH of 35.5% (Martin et al., 2017) and the cluster average $\kappa$ given in Table 1. 
[revised manuscript text omitted]

*We would also like to thank the two anonymous peer reviewers for their helpful suggestions and insights.*

**Tables**

**Table 1** **Aerosol and meteorological parameters for each of the cluster time periods,  with the total number and percentage of observations in each cluster.** Cluster mean values are given for total particle number concentration, κ hygroscopicity parameter from the srCCN system, HYSPLIT 24-hour accumulated precipitation along the trajectory, and local wind velocity observations.

| BML Cluster | Percentage of Observations % (total N) | Total Number Concentration (# cm$^{-3}$) (std dev) | κ Mean (-) (std dev) | Wind Velocity u (m/s) | v (m/s) | Trajectory 24-hr Accum Precip (mm) (std dev) |
|---|---|---|---|---|---|---|
| M1 | 5% (169) | 592 (435) | 0.49 (0.23) | 6.76 | -4.22 | 0.31 (0.73) |
| M2 | 9% (308) | 774 (462) | 0.30 (0.10) | 2.77 | 1.16 | 0.07 (0.30) |
| M3 | 10% (345) | 1547 (1177) | 0.46 (0.22) | -2.63 | 5.70 | 1.02 (1.76) |
| T1 | 9% (304) | 1975 (2166) | 0.25 (0.17) | 0.25 | 4.12 | 0.51 (1.21) |
| T2 | 17% (566) | 3189 (2055) | 0.18 (0.07) | -1.90 | 2.70 | 0.16 (0.70) |
| T3 | 16% (545) | 3816 (3447) | 0.15 (0.06) | -0.70 | 2.25 | 0.03 (0.21) |
| T4 | 20% (659) | 2306 (1166) | 0.17 (0.06) | 0.46 | 2.49 | 0.01 (0.04) |
| T5 | 14% (461) | 1459 (600) | 0.20 (0.07) | 2.72 | 0.67 | 0.00 (0.00) |

**Table 2** **Cluster best-fit number size distribution and activated fraction parameters are shown, with activated fraction parameters a, b, and c pertaining to the fit model equation given in the supplemental information. Clusters with "M" and "T" names refer to marine and terrestrial aerosol population types, respectively.**

[revised manuscript text omitted]